# A Novel Benchmark Framework for Neural Embeddings in Earth Observation

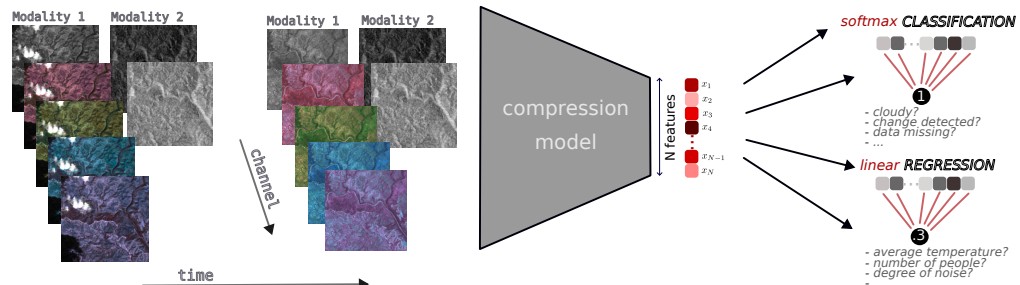

Cartoon summary of the benchmark framework. Multi-temporal, multi-modal, and multi-channel inputs $x$ are compressed into fixed-size embeddings $z = E(x)$ by a user-defined encoder $E$. The embeddings are then linearly probed on a diverse set of regression and classification downstream tasks to assess the *general-purpose* quality of $z$.

## Abstract

We introduce a novel benchmark framework for evaluating (lossy) neural compression and representation learning in the context of Earth Observation (EO). Our approach builds on fixed-size embeddings that act as compact, task-agnostic representations applicable to a broad range of downstream tasks. Our benchmark comprises three core components: (i) an evaluation pipeline built around reusable embeddings, (ii) a new challenge mode with a hidden-task leaderboard designed to mitigate pretraining bias, and (iii) a scoring system that balances accuracy and stability. To support reproducibility, we release a curated multispectral, multitemporal EO dataset. We present initial results from a public challenge at a workshop and conduct ablations with state-of-the-art foundation models. Our benchmark provides a first step towards community-driven, standardized evaluation of neural embeddings for EO and beyond.

## 1 Introduction

The rapid growth of visual data, from online media to scientific observation, has made efficient compression a central challenge for storage, transmission, and large-scale analysis (Pouyanfar et al., 2018; Wang et al., 2018a; Gomes et al., 2025). Traditional codecs such as JPEG2000 (Skodras et al., 2001) and more recent learned autoencoders (Ballé et al., 2016) are optimized for pixel-level distortion, largely reflecting human visual perception. However, many machine learning pipelines care less about perceptual fidelity and more about semantic fidelity, retaining the information needed to solve downstream tasks (Huang and Wu, 2024). This gap is particularly critical in domains like Earth Observation (EO), where petabyte-scale datasets of multi-modal satellite imagery must support diverse analytical tasks ranging from environmental monitoring to disaster response (Guo et al., 2017). EO data are characterized by substantial redundancy and noise across multiple spectral bands and temporal sequences, amplifying the need for compression strategies that efficiently capture underlying, task-relevant information (Gomes et al., 2025). This gives rise to the question: *How much task-relevant information can be squeezed into compact data representations?*

Recent work has shown that compressed latent representations can preserve rich semantic content, enabling pipelines to operate directly on features without reconstructing the input image (Torfason et al., 2018; Singh et al., 2020). Self-supervised foundation models (FMs) further demonstrate that embeddings can transfer across tasks with minimal fine-tuning. Yet, their dimensionality often rivals or exceeds the size of the original data, reintroducing storage and bandwidth bottlenecks (Gomes and Brunschwiler, 2024; Lu et al., 2024). Despite these advances, there is currently no standardized framework evaluating how effectively compressed representations retain semantic content across multiple downstream tasks. Existing evaluations remain fragmented, often restricted to pixel fidelity, single-task utility, or unconstrained high-dimensional embeddings, making it challenging to compare approaches on a common basis.

To address this, we introduce a model-agnostic benchmark for assessing the semantic quality of embeddings in EO. Our frameworks is designed to (1) evaluate compressed embeddings under strict size constraints, (2) probe semantic retention using linear models across diverse downstream tasks, (3) support multi-modal and multi-temporal data typical of data-intensive EO settings, and (4) foster community contributions, including new datasets and compressors—towards establishing open, task-centric compression standards. Our key contributions are:

- Section 3 – **Benchmarking Framework:** We develop a standardized framework for evaluating compressed embeddings via downstream tasks, aligning with task-centric machine-to-machine workflows.
- Section 4 – **Benchmark Tasks:** We curate and release a suite of novel EO downstream tasks, spanning cloud analysis, agricultural monitoring, forest quantification, urban heat islands identification, and land cover analysis.
- Section 5 – **Benchmark Evaluation:** We validate the utility of our benchmark through a data challenge, introducing a novel hidden-task evaluation scheme. We further test embedding quality under diverse compression strategies, including pre-trained neural compressors and FMs.

## 2 RELATED WORK

Below, we review research fields relevant to contextualize our benchmark framework:

**Classical rate-distortion compression.** Image and video codecs such as JPEG, JPEG2000, H.264/HEVC (Wallace, 1991; Skodras et al., 2001; Sullivan et al., 2012; Richardson, 2010) exploit handcrafted transforms (Goyal, 2001; Bracewell, 1986; Daubechies, 1992) and entropy coding to reduce statistical redundancy. Their performance is evaluated through the rate–distortion (RD) trade-off between compressed bit rate and reconstruction fidelity (e.g., MSE, PSNR).

**Neural image compression.** Learned autoencoders replace handcrafted transforms with analysis and synthesis networks jointly optimized for rate and distortion. Differentiable entropy models enable superior RD performance compared to JPEG2000 (Ballé et al., 2016; Theis et al., 2022), with subsequent extensions using hyperpriors (Ballé et al., 2018; Minnen et al., 2018), autoregressive models (Minnen and Singh, 2020), and transformers (Qian et al., 2022). With automated vision pipelines, the concept of compression for machines shifts focus from human-perceptual fidelity to task-driven utility. End-to-end approaches jointly optimize compressors with task networks (Chamain et al., 2020; 2021; Le et al., 2021; Codevilla et al., 2021; Wang et al., 2021; 2023a; Fischer et al., 2025). Other methods enforce invariance to task-relevant augmentations through self-supervised objectives (Dubois et al., 2022), or bypass reconstruction by training tasks directly on compressed latents (Torfason et al., 2018; Duan et al., 2023; Singh et al., 2020).

**Compression in EO.** EO imagery presents unique compression challenges, with multi-spectral bands, temporal sequences, and petabyte-scale archives (Guo et al., 2017; Wilkinson et al., 2024). Traditional pipelines often rely on codecs like JPEG2000 (Yeh et al., 2005). Recent neural approaches extend rate-distortion autoencoders to EO imagery, achieving significant rate-distortion improvements on multispectral data (Alves de Oliveira et al., 2021; Kong et al., 2021; Cao et al., 2022), while temporal compression remains underexplored (Du et al., 2024; Wang et al., 2018b). For a comprehensive review, see (Gomes et al., 2025). Importantly, most works evaluate RD, not task relevance.

**Implicit neural representations.** INRs have recently emerged as a compelling alternative for compactly encoding Earth observation data. INR-based approaches have been explored for global location embeddings from satellite imagery (Klemmer et al., 2025), hyperspectral compression using neural radiance fields (Zhang et al., 2024; Rezasoltani and Qureshi, 2024), and remote sensing image compression via coordinate-based networks (Li et al., 2023). More generally, INRs have shown strong potential for image compression and continuous signal representation (Strümpler et al., 2022; Dupont et al., 2021; Sitzmann et al., 2020). While these methods are outside the scope of our current baseline evaluation, they represent a promising direction for future extensions of our benchmark.

**EO Foundation Models.** Self-supervised learning has enabled large-scale vision foundation models (FMs) pretrained on vast, unlabeled satellite datasets using masked reconstruction, contrastive, or predictive tasks (Wang et al., 2022a; Sun et al., 2022; Wang et al., 2022b; Mai et al., 2022; Wang et al., 2023b; Hong et al., 2023; Jakubik et al., 2023; Liu et al., 2024). These FMs produce versatile high-dimensional embeddings for EO downstream applications, such as flood segmentation, land-use mapping, and environmental monitoring. More recently, multimodal EO foundation models have begun fusing data modalities, such as SAR and optical imagery, to capture diverse geophysical

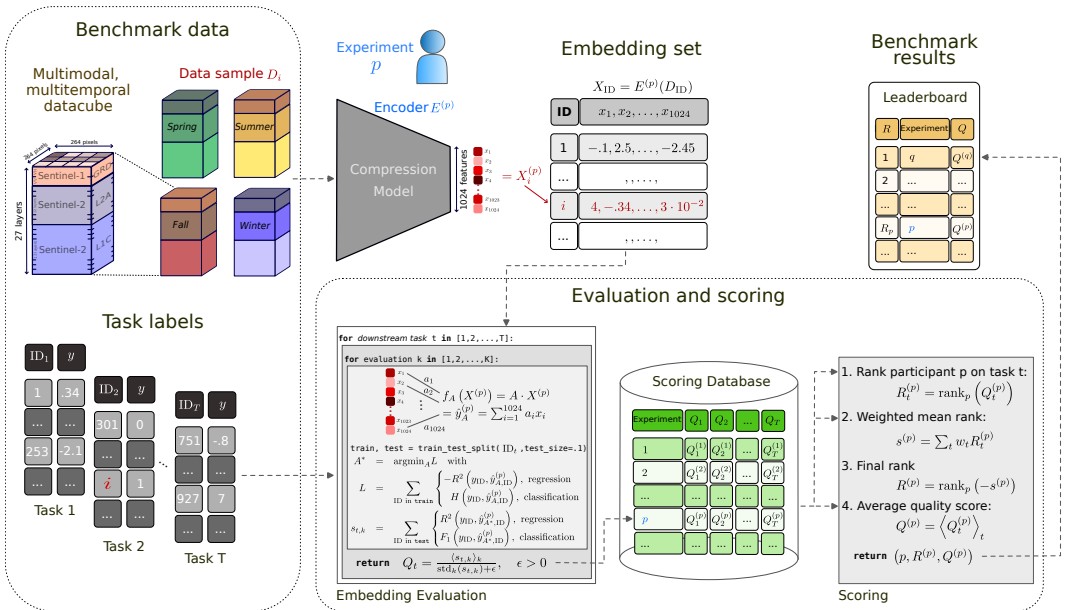

Figure 1: Workflow diagram: The user of our benchmark compresses a set of downstream data into fixed-length embeddings (size $N = 1024$ here). Our benchmark loads the embeddings for each of the $T$ downstream tasks, performs $K$ evaluations, randomly sampling a training and test split from the task data each evaluation, and scores the result compared to all previous experiments as stored in the *Scoring Database*.

characteristics and improve application performance (Li et al., 2022; Fuller et al., 2023; Xiong et al., 2024; Wang et al., 2025; Jakubik et al., 2025; Brown et al., 2025). However, with the exception of (Brown et al., 2025), the resulting latent representations often rival or exceed the original data size, creating data transfer and data processing bottlenecks. Gomes and Brunschwiler (2024) addresses these challenges by integrating neural compression into FM bottlenecks. On image level, Rolf et al. (2021) utilizes fixed, random convolutional kernels to engineer features as basis for linear regression to predict user labels.

**EO benchmarks.** Current EO domain benchmarks, such as GEO-Bench (Lacoste et al., 2023) and PANGAEA (Marsocci et al., 2025), evaluate FMs by fine-tuning backbones or training complex decoders on intermediate features. These approaches typically require model access and significant computational resources, with limited consideration given to factors such as embedding size and workflow efficiency. In contrast, our benchmark evaluates fixed-size embeddings through task-agnostic linear probing without any need for access to model backbones. In fact, our approach treats the encoder as a black box that converts any input to a given number of features. Our benchmark provides a lightweight, size-aware evaluation protocol for efficient local testing, and it is structured as a flexible, extendable framework, designed to accommodate future downstream tasks (e.g., *Copernicus-Bench* in Wang et al. (2025)) and evaluation methods. Moreover, our benchmark can be deployed as a novel challenge format that simulates real-world scenarios by requiring participants to submit compressed EO embeddings without prior knowledge of the specific downstream tasks. This setup reflects the demand for broadly generalizable embeddings. As demonstrated in our data challenge, our benchmark integrates with established platforms such as EvalAI (Yadav et al.), and is designed to support future competitions on new, unseen tasks.

## 3 BENCHMARKING FRAMEWORK

At the heart of our benchmark framework resides (i) an embedding evaluation workflow and (ii) a ranking method to fairly compare performance across multiple tasks of varying difficulty.

**Evaluation workflow.** Figure 1 visualizes the pipeline for an *Experiment* $p$ compressing the samples indexed by $i$ of *Benchmark data* to create a set of fixed-size embeddings (*Embedding set*) through an *Encoder* $E^{(p)}$: These embeddings $X^{(p)}$ are provided to our benchmark, which performs the evaluation given corresponding *Task labels* (aka *downstream task*s $t = 1 \dots T$) to return the *Benchmark results*

through a *Leaderboard*. For each *Experiment*, our benchmark framework performs an *Embedding Evaluation* given *Multimodal, multitemporal datacubes* across a set of downstream tasks undisclosed to the developers of a given *Compression Model* comprising an *Experiment* $p$. Correspondingly, our framework aggregates scores $s_{t,k}$ per *training and test split* $k$ to gather statistics for the quality score $Q_t$ per downstream task $t$. Consequently, a *Scoring* algorithm applies a task difficulty-dependent ranking scheme.

**Evaluating embeddings.** For our benchmark, each input sample must be represented as a fixed-size embedding to compress an input data cube, e.g., our EO downstream tasks as detailed in Section 4, or future extensions, cf. Section 6. Currently, image-level linear regression and binary (softmax) classification are supported. Our benchmark enforces embeddings of fixed, but configurable, size but otherwise does not constrain how embeddings are generated. Following Fig. 1, our benchmark evaluates the compressed embeddings $X^{(p)}$ of an experiment $p$ as follows: For each task $t = 1 \dots T$, $K$ linear classifiers (with $N$ tunable parameters $a_{1 \dots N}$ plus bias term $a_0$) are trained to fit the downstream task labels $y_{\text{ID}}$. Each $k = 1 \dots K$ denotes a separate, randomly generated `split` of the downstream task $t$ into a `training` and `test`ing set. For each tuple $(t, k)$ our benchmark computes an accuracy measure $s_{t,k}$, utilizing $R^2$ (R-squared) for regression tasks and the $F_1$ score for (binary) classification. From the set $\{s_{t,k}\}_{k=1\dots K}$, our benchmark derives a signal-to-noise-like *quality score* $Q_t^{(p)}$ as the mean performance on task $t$ sensitive to the variability in performance of experiment $p$:

$$Q_t^{(p)} = 100\epsilon \frac{\langle s_{t,k}\rangle_k}{\text{std}_k\left(s_{t,k}\right) + \epsilon}. \tag{1}$$

Here, $\langle \cdot \rangle_k$ denotes averaging and $\text{std}_k\left(\cdot\right)$ the standard deviation as calculated over the $K$ splits. The parameter $\epsilon > 0$ acts as a regulator avoiding high variability in $Q_t^{(p)}$ for small $\text{std}_k\left(s_{t,k}\right)$. The quality score Eq. (1) varies in $[0, 100]$ for both, classification and regression. Thus, $Q_t^{(p)}$ allows for an interpretation of mean accuracy in percent. Compared to using the mean $R^2$ over the $K$ splits, $Q_t^{(p)}$ penalizes methods with larger variance in the $R^2$. Further details on the quality score is provided in Section A.1 of the supplementary material. We note: experiments that perform worse than simply predicting the mean of labels $y_{\text{ID}}$ for regression tasks result in negative $s_{t,k}$ degrading the mean performance. In fact, compression models with negative $Q_t^{(p)}$ should be flagged unreliable—they seriously underperform.

**Task difficulty-dependent ranking.** A novel scoring method is introduced by our benchmark. It is designed to compare the overall performance of multiple participants over multiple tasks. Based on a rank-then-aggregate approach (Wiesenfarth et al., 2021), our benchmark dynamically weights the performance across tasks depending on their relative difficulty: Each experiment initially receives a rank $R_t^{(p)}$ per task, with the best rank given to the experiment with highest $Q_t^{(p)}$. To break ties, all tied experiments are given the lower (better) rank. An experiment's final rank is calculated from the weighted mean rank across all tasks:

$$s^{(p)} = \sum_{t=1}^{T} w_t R_t^{(p)} \quad \text{with} \quad w_t = \text{std}_p\left(Q_t^{(p)}\right) \Big/ \sum_{t=1}^{T} \text{std}_p\left(Q_t^{(p)}\right) \tag{2}$$

where the tasks are weighted by the standard deviation of the $Q_t^{(p)}$ of all experiments on the task. The weighting scales the importance of the tasks such that (a) tasks where all participants perform similarly receive low importance, and (b) tasks where the participants differentiate between each other are weighted highly. Our benchmark also provides the *mean Q* value $\langle Q_t^{(p)}\rangle_t$ as an experiment-specific measure of performance. For scenarios with few experiments where the interpretation of a ranking is limited in terms of task difficulty, $\langle Q_t^{(p)}\rangle_t$ serves as an alternative metric to compare (individual) experiments. Based on the setup/*mode of operation*, $s^{(p)}$ or $Q_t^{(p)}$ may be preferred. The former serves competitive challenge settings with novel downstream tasks of unknown levels in difficulty. The latter is favorable for long-term leaderboards where re-ranking of all methods should not depend on the update of a single method. Section A.1 provides additional analysis of the ranking scheme.

## 4 BENCHMARK TASKS

Our benchmark provides a set of pre-processed, heterogeneous downstream tasks designed for continuous extension in the future. The initial release provides regression labels that are easily turned into binary classification tasks through a threshold.

Table 1: Summary of spatial coverage for current set of data cubes and associated number of downstream tasks.

| Dataset | Spatial Coverage | Temporal Coverage | Years of Labels | # Samples | # Tasks |
|---------|------------------|-------------------|-----------------|-----------|---------|
| Crops | US Corn Belt | 2022 | 2023[1] | 3355 | 1 |
| Landcover | Europe | 2018 | 2018 | 4691 | 2 |
| Biomass | Global | 2019 | 2019 | 2415 | 2 |
| Clouds | Global | 2018 - 2020 | 2018 - 2020 | 1140 | 1 |
| Heatisland | Northern Hemisphere | 2022 | 2021 - 2024 | 1659 | 2 |

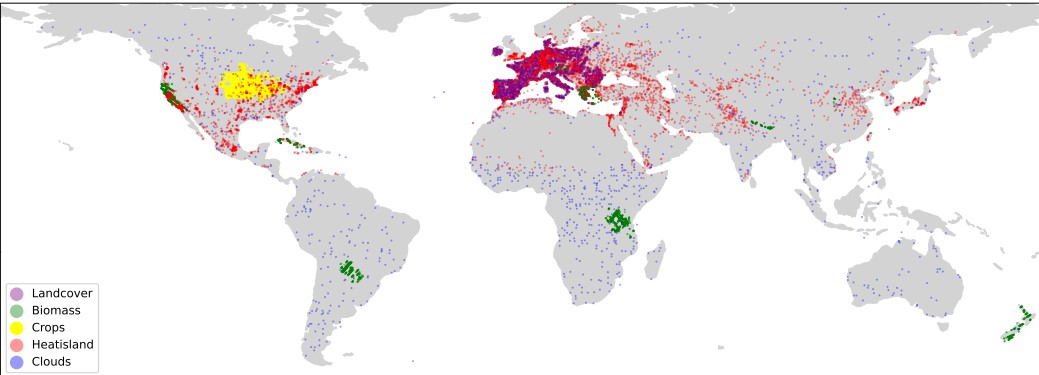

Figure 2: Spatial distribution of the downstream tasks.

We utilize 13 channels of Sentinel-2 Level-1C *Top-of-the-Atmosphere* (S2L1C) and 12 channels of multi-spectral Sentinel-2 Level-2A *surface reflectance* (S2L2A). On top we spatially align 2 channels of radar Sentinel-1 (S1) GRD product polarizations (VV and VH). For a given geolocation we retrieve four timestamps, one per season: winter (Dec–Feb), spring (Mar–May), summer (Jun–Aug), and fall (Sep–Nov). Figure 1 depicts these four seasonal data cubes, with each containing 27 bands. Google Earth Engine (Gorelick et al., 2017; GEE) (GEE) was utilized to download all relevant satellite data. All labels, except for the Clouds use case, have been retrieved from GEE, too. The processed data was stored as cloud-ready ZIP-Store of the Zarr file format.

The downstream tasks contain between 1100 and 4691 samples (locations/labels), which are distributed globally (Fig. 2). The associated satellite data cubes are pre-processed and filtered to ensure UTM-projected patches with a size of 264 × 264 pixels without spatial overlap.

The **crops** task covers cropland in the US Corn Belt and is provided by the US Department of Agriculture (USDA) (Boryan et al., 2011). Soybean and corn were selected as primary focus classes for this downstream task, with the fraction of corn and soybean within each patch serving as the label. The Crop Data Layer is published annually with a spatial resolution of 30 meters. The labels are available as post-processed data[1].

The **landcover** tasks leverages aggregated land use data from the European Environment Agency (EEA), which includes various land cover classes such as forests, urban areas, water areas, and agricultural land (European Environment Agency (EEA), 2018). These labels represent the dominant land cover within each patch at a spatial resolution of 100 meters within Europe. Based on this data, two downstream tasks are provided for forests and agricultural land 2018 within the challenge.

The **biomass** tasks uses above-ground biomass estimates derived from LIDAR measurements from the Global Ecosystem Dynamics Investigation (GEDI) instrument. GEDI provides structural information on vegetation height and density, allowing robust models to estimate above-ground biomass in megagrams per hectare (Mg/ha) (Dubayah et al., 2022). Within our benchmark, the GEDI Level 4A biomass estimates were spatially aggregated to the satellite patches with 264x264 pixels, providing a mean biomass value and its standard deviation as regression targets.

The **clouds** provides cloud cover fractions based on CloudSen12+ (Aybar et al., 2024) as labels and pre-processed Sentinel-1 and Sentinel-2 data cubes as corresponding observations. Although the

---

[1] e.g., the label year 2023 corresponds to crops cultivated during the 2022 growing season

SAR data is not affected by clouds, Sentinel-1 is included alongside Sentinel-2 to ensure a consistent data structure for all downstream tasks.

For the **heatisland** use case, Landsat-8 Land Surface Temperature (LST) provides surface temperature data that are used as labels for urban areas (Observation and Center, 2020). This is particularly relevant in the context of heat events and future urban planning, and contains 1659 samples. The corresponding tasks address the mean surface temperature and its standard deviation per data cube stack. Further information on the data and their downstream tasks during the competition can be found in the Section B.1.

## 5 BENCHMARK EVALUATION

We describe how we tested our benchmark in a real-world setting, by detailing the experimental setup in Section 5.1, as well as discussion of outcomes and learnings in Section 5.2. We further present baseline evaluations exploring our set of downstream tasks in Section 5.3.

### 5.1 DATA CHALLENGE VALIDATION

To validate our benchmark under realistic conditions, we utilized it in a data challenge. Participants were tasked with compressing multi-modal, multi-temporal EO imagery, cf. Section 4, into 1,024-dimensional embeddings. Given the benchmark input data cubes, this amounts for a compression ratio of approx. 7,000. Crucially, participants did not know which or the number of downstream tasks their embeddings would be evaluated on; this hidden-task design discourages overfitting and encourages the development of general-purpose EO representations. Participants were ranked, according to Section 3, across two sets of downstream tasks. One modification was made to the dataset compared to the dataset described in Section 4; the clouds task targets were mainly replaced by zeros, causing heavy skewness in the labels and basically random connection between the imagery and labels.

**Phases.** In a three-week development phase, teams developed embedding methods using a publicly available dataset. A partial release of 5 tasks, each with a subset of its samples but no information of the task type, allowed participants to receive initial feedback for development. Submissions returned only the mean $Q$ value to prevent leakage of task-specific performance and information.

In the subsequent three-day evaluation phase, an extended set of 9 tasks, and new data on the tasks also used in the development phase, was released. The teams had three days and up to three submissions to encode and submit embeddings; these runs defined the final leaderboard standings. By the end, two winning teams were chosen; The first based on the dynamic ranking scheme, and the other as the team with highest mean $Q$ score.

**Platform and infrastructure.** The benchmark framework was modified to utilize Eval.AI (Yadav et al.) to collect submissions. Our benchmark ran on a separate 8-vCPU server, retrieved new submissions via API, executed the evaluation, and pushed results to a custom leaderboard hosted on GitHub, displaying the dynamic ranking described in Section 3, as well as back to Eval.AI. Additional details are available in Section A.2.

### 5.2 DATA CHALLENGE RESULTS

**Participation and ranking.** Twenty-three teams submitted to the development phase; sixteen went on to the final evaluation, nine of which shared their submissions publicly. The quality scores $Q_t^{(p)}$, shown in Fig. 3 for the evaluation phase, varied widely from 0–5 on some tasks to 5–40 on others. The evaluation method, cf. Eqs. (2) to (4) and Section A.1, efficiently scaled task importance ensuring that the tasks impacted the leaderboard relative to their differentiating effect across the teams. Notably, the weighting reduced the impact of the tasks with random labels. Further, the dynamic ranking caused a swap between the original first and second place methods due to a third team, highlighting the impact of adaptive weighting.

**Top methods.** The team that achieved the best overall rank and the team with the highest average $Q$-Score both built their embeddings by ensembling multiple FM representations, although with different approaches; One pre-training backbones, and the other training a bottleneck based on frozen FM backbones. The fourth-place team took a different path, forgoing any pre-training and instead generating embeddings using the MOSAIKS method (Rolf et al., 2021).

**Key takeaways.** Running our data challenge demonstrated that our benchmark efficiently evaluates and ranks the performance of compact embeddings over multiple downstream tasks. The scoring method produced a more balanced and discriminative ranking compared to uniformly weighting

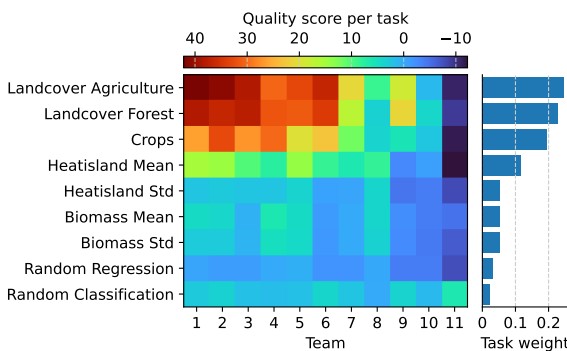

Figure 3: Quality score $Q_t^{(p)}$ of the participants of the data challenge evaluation phase with corresponding task weight used for ranking. Teams are ordered by their final leaderboard rank, with the winner as Team 1. Team 10 is a simple averaging baseline described in Section 5.3 and 11 denotes a baseline of normally distributed random embeddings. We observe: In our setup the landcover tasks have been the most discriminative to rank models. The random (control) task was the most "difficult" as all models failed (by design). From the analysis of results, estimating biomass seems a challenge for real-world downstream tasks.

the tasks, particularly noticeable in the down-weighting of the two random tasks. Hiding the tasks efficiently prevented overfitting, ensuring fairness between the participants. Both winning solutions were based on FMs, indicating these can indeed provide semantically rich, general embeddings. However, also non-FM based solutions scored high.

## 5.3 GENERAL EVALUATIONS

We assess our framework through a series of experiments, including embeddings from self-supervised FMs and representations from learned neural compressors. Given our compression requirements, we apply spatial and temporal aggregation techniques to obtain the required compact embedding sizes.

Our analyses span three perspectives: First, we follow the challenge setup and constrain all methods to produce 1,024-dimensional embeddings. Second, we relax this requirement to study how embedding dimensionality affects downstream performance. By varying the size of FM-based embeddings, we examine how larger or smaller representations influence performance, motivating the challenge-default of 1,024 dimensions. Lastly, we revisit and explore the assumption of linear probes as decoder models.

**Embedding Aggregation Methods.** Each input consists of four seasonal snapshots from Sentinel-1 (radar) and Sentinel-2 (optical, w/ and w/o atmospheric correction). We benchmark image-based encoders by applying temporal averaging across the four timesteps, either *before* encoding (pre-encoding aggregation) or *after* encoding (post-encoding aggregation). For spatial aggregation, convolutional encoder outputs are reduced through spatial averaging (pooling) in the embedding space, with additional pairwise channel means applied when further dimensionality reduction is needed. For ViT encoders, we average the spatial patch tokens to obtain a single embedding. CLS-token evaluations, together with an extended suite of baseline methods, is provided in Section B.3. The aggregated embeddings are zero-padded into the expected embedding dimension. To study the role of input modalities, we evaluate unimodal encoders that use only S2L1C bands and multimodal encoders that process the full SSL4EO-S12 data cube.

**Encoder Baselines.** A simple **averaging baseline**, which applies spatial pooling, channel-wise averaging, and flattening serves as a minimal reference point, see Section B.3.2 for implementation details. **Neural rate–distortion compressors** are implemented via Factorized Prior autoencoders (Ballé et al., 2016) pretrained on S2L1C data. We extract latent bottleneck features before entropy coding (cf. (Torfason et al., 2018)) and aggregate as described above. Finally, we benchmark publicly available EO **FMs**, including ResNet (He et al., 2015) and ViT (Dosovitskiy et al., 2020) backbones pretrained with masked-autoencoding (MAE (He et al., 2021)) or contrastive (DINO (Caron et al., 2021)) objectives, as well as the multimodal TerraMind model (Jakubik et al., 2025), which is pretrained jointly on radar and optical inputs.

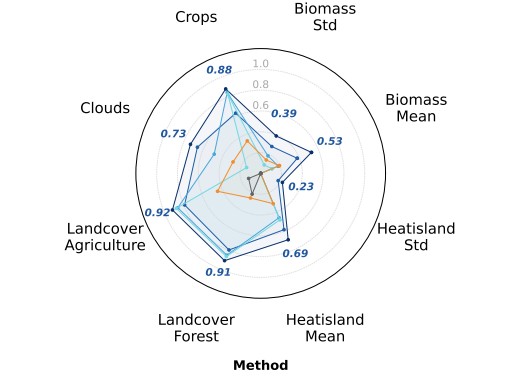

| Method | Pretraining | Loss | Backbone | Input | Downsampling | Original latent dim. |
|--------|-------------|------|----------|-------|--------------|----------------------|
| TerraMind (FM) | TerraMesh | Cross-Entropy | Multi-modal ViT-B/16 | All (S1,S2) | Patch averaging | $196 \times 768$ |
| MAE ViT (FM) | SSL4EO | MAE | ViT-B/16 | S2L1C | Patch averaging | $196 \times 768$ |
| DINO ViT (FM) | SSL4EO | DINO | ViT-B/16 | S2L1C | Patch averaging | $196 \times 768$ |
| DINO ResNet (FM) | SSL4EO | DINO | ResNet-50 | S2L1C | Global avg., pairwise channel mean | $2048 \times 7 \times 7$ |
| Factorized Prior | SSL4EO | Rate-Distortion | CNN | S2L1C | Adaptive pool ($4 \times 4$), pairwise channel mean | $128 \times 14 \times 14$ |
| Averaging Baseline | – | – | – | All (S1,S2) | bilinear avg., flatten | $4 \times 4 \times 8 \times 8$ |

Figure 4: (Left) Per-task $R^2$ performance for a representative subset of embedding–compression methods. Each axis corresponds to one downstream prediction task; the center denotes $R^2 = 0$, negative values are clipped to zero for clarity. (Right) Summary of method configurations.

**Results and discussion.** Figure 4 presents the linear-probing performance in terms of $R^2$ for all downstream tasks assembled and a subset of tested baselines, chosen for clarity and representation. Descriptions and results for all evaluated methods are provided in Section B.3.2 of the supplementary material. Embeddings from our neural rate–distortion compressors outperform the simple averaging baseline but remain below $R^2 = 0.5$. This observation highlights the characteristics of our setup with high compression rate $\sim$7,000 while testing under linear probing. FM embeddings reveal a task-dependent trend: FMs in general, with contrastive (DINO) and multimodal models (TerraMind) in particular, achieve high $R^2$ on semantic tasks where multi-pixel context is relevant (e.g., land-cover proportion). However, certain FMs struggle on geophysical predictions of quantities resolved at the sub-pixel level (e.g., biomass estimation), whereas multimodal and MAE FMs strike a better balance across tasks.

**Temporal aggregation.** Across all methods, *post-encoding aggregation* consistently outperforms *pre-encoding aggregation*. Accordingly, the results in Figure 4 employ post-encoding aggregation exclusively, despite incurring a $\times 4$ increase in embedding-generation runtime. The performance gains are modest for static-feature tasks (e.g., land-cover), yet substantial for temporally sensitive tasks (e.g., cloud-fraction estimation), underscoring the importance of preserving per-snapshot details.

**Embedding size.** In Fig. 5a we numerically study how the embedding size impacts performance testing the models introduced above. Given our setup we distilled these empirical observations:

- **CNN backbones.** Performance peaks for embedding sizes $128 \lesssim N \lesssim 1024$, with accuracy dropping outside. Larger embeddings add computational cost without significant gain in performance.
- **ViT backbones.** We find the best performance at $N = 1024$, the upper limit allowed by the embedding dimension. A lower $N$ consistently reduce accuracy—except for certain regression tasks such as for *Biomass*.
- **Trade-offs.** While larger embeddings increase the number $N = |A|$ of (linear) probe parameters $A$ (cf. Fig. 1), smaller $N$ often fail to retain task-relevant semantics.

Our experiments support an embedding size of $N = 1024$ as a balanced default across downstream tasks, cf. Fig. 2. The benchmark framework is flexible to explore size–utility trade-offs, analogous to rate–distortion analysis in neural compression. Section B.3.3 provides detailed per-task performance plots.

**Linear probing assumption.** We evaluate embedding quality using linear probing, a widely adopted practice in representation learning (Xu and Tewari, 2021) to focus on embeddings without fine-tuning encoder backbones. While non-linear probing (e.g., small MLP heads) can in principle capture richer structures, it risks compensating for poor embedding quality Plachouras et al. (2025). Our experiments in Fig. 5b and Section B.3.3 demonstrate: Replacing linear probing by small, non-linear decoders yields only marginal gains for top-performing embeddings, while providing larger improvements for weaker ones. Further, non-linear probing substantially increases computational cost. Thus, linear probing remains an (energy-)efficient and reliable measure of how much semantically relevant

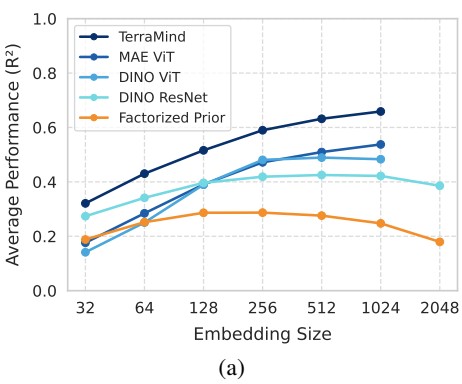 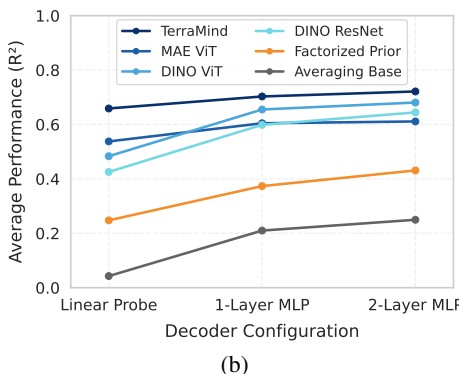

|                  |                  |
|:----------------:|:----------------:|
| (a)              | (b)              |

Figure 5: (a) Average downstream $R^2$ as a function of embedding size $\log N$. Largest size is the full channel count (CNN) or native patch-token dimension (ViT). (b) Average downstream $R^2$ comparing linear probing against one-layer and two-layer MLP probes on 1024-dimensional embeddings.

information is directly accessible from an embedding space. Linear probing efficiency enabled our benchmark to run over 400 submissions within minutes on commodity hardware.

## 6 FUTURE WORK

**Reproducibility.** Data contributions to our benchmark demands a permissive CC-BY 4.0 license. Our data challenge required the winners to release their solution under Apache 2.0 license[2]. The same holds for any future extension of our benchmark where Section B.5 provides further details. While Section A.1 has background on the theoretical basis of our evaluation metric with references to code, Sections A.2 and B.4 share specifics on the hardware, the software environment, and hyperparameter settings our benchmark runs on. Section A.3 provides some additional elements of running the data challenge. Sections B.1 and B.2 gathers general facts on the benchmark data and code framework, respectively. Section B.3.2 is dedicated to additional model performance metrics for downstream tasks provided by our benchmark.

**Fixed-size compression.** Our current evaluation emphasizes fixed-size embeddings, as fixed-size vectors enable fast retrieval, comparison, and inference, critical for machine-oriented downstream tasks. Nevertheless, the framework can be naturally extended to incorporate entropy coding, where embeddings are further losslessly compressed for transmission before decoding and use. In this setting, the proposed performance scores directly evaluate utility as a function of the final entropy-coded bitrate, thereby bridging task performance with classical rate–distortion analysis.

**Choice of tasks.** The discriminative power of results depends on downstream tasks. We curated diverse, image-level tasks focused on global semantic content, together with a dynamic ranking scheme. Future extensions will include spatially structured tasks such as pixel-level segmentation or time-sensitive predictions, which may require less aggressive compression ratios than the current value of ~7,000. Probing strategies may also evolve as tasks grow in complexity.

**Downstream data.** Although the current benchmark is rooted in EO, its design is domain-agnostic. Extensions can cover multi-modal, spatio-temporal data across domains such as weather forecasting, medical imaging, or autonomous driving. Our experiments leveraged SSL4EO-S12 as an initial sweet spot (multi-modal, multi-temporal, multi-spectral). The same concept readily transfers along the axes of *domain*, *modality*, *time*, and *channels*.

**Building a community.** With our benchmark, we provide a seed to grow an eco-system centered around benchmarking highly-compressed embeddings on a set of standardized, community-contributed downstream tasks. To avoid contributing to an ever-growing number of benchmark datasets, we intend to harmonize with existing ones, such as GEO-Bench and PANGAEA. Our benchmark's mission is to provide a standardized framework of benchmarking embeddings. We warmly welcome contributions from all research areas involved in neural compression. Future extensions of the benchmark framework include pixel-wise and temporal downstream tasks.

---

[2] https://creativecommons.org/licenses/by/4.0 and https://www.apache.org/licenses/LICENSE-2.0.html

## 7 CONCLUSION

We presented the first of its kind, task-driven benchmarking framework for compression that evaluates neural embeddings by downstream task performance, rather than pixel fidelity. The framework introduces a novel rank-then-aggregate scoring method which dynamically determines the task complexity based on score statistics. We demonstrated our benchmark framework by setting up a data challenge on multi-modal, multi-temporal, and open-source EO data. We introduced a novel set of real-world downstream tasks which remained undisclosed at the time of the data challenge, and have been publicly released after the conclusion of the competition. Our benchmark encourages the development of methods that generate semantically-rich, general-purpose embeddings.

For our setup, experiments demonstrated that multi-modal foundation models yields strong overall performance—particularly on semantic land-cover tasks. Post-encoding fusion of seasonal views resulted in notable gains for temporally sensitive tasks such as cloud cover prediction. We also observed that smaller and, in some cases, larger embedding sizes may degrade performance. This observation highlights compact embeddings as a practical choice for image-level tasks when high-quality annotations and compute resource become scarce.

Our benchmark is open source and ready for extension—either by novel evaluation methods or additional downstream tasks without any conceptual restriction to Earth observation. Currently, our benchmark framework is limited to image-level tasks, but future work aims to extend our benchmark's functionalities to include pixel-wise outputs, options beyond linear probing, and an assessments of bit-rate efficiency.

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

# A   TECHNICAL DETAILS OF THE DATA CHALLENGE

Before we delve into generic considerations regarding our benchmark in Section B, we introduce its origin spared by the innocent question:

> *If Geospatial Foundation Models claim to generate informative, generic feature vectors for a broad range of use cases, why can't we put that claim to the test in a data challenge? Catch: We will not disclose the downstream tasks, but simply ask to embed /compress Earth observation data.*

## A.1   CHALLENGE EVALUATION METHOD AND CONFIGURATION

**An individual, *local* score of embedding quality.** As detailed in the main article, the central evaluation metric serving as quality score to answer the question above reads like

$$Q_t^{(p)} = 100\epsilon \frac{\langle s_{t,k}\rangle_k}{\text{std}_k\left(s_{t,k}\right) + \epsilon} \equiv Q = 100\epsilon \frac{\bar{s}}{\Delta s + \epsilon} \quad \text{with} \quad \epsilon = 0.02 \quad . \tag{3}$$

For fixed $\epsilon > 0$, the maximum value of $Q$ reaches $100\epsilon\epsilon^{-1}\bar{s} = 100\bar{s}$ when the statistical fluctuations vanish, $\Delta s \to 0$. Given $\Delta s \geq 0$ and $\bar{s} \in [0,1]^3$ derived from a measure such as the F1-score or R-squared, the range of the quality score can be interpreted as a *percentage of quality*.

The numerical value of the regulator $\epsilon$ determines the scale at which $Q$ becomes insensitive to statistical fluctuations $\Delta s$: As long as $\Delta s \gg \epsilon$, in zero-order approximation, we have $q = Q/100 \approx \bar{s}/\Delta s$ a measure of signal-over-noise for the quantity $s$. At the other end of the spectrum when $\epsilon \gg \Delta s$ dominates the noise, we conclude $q \approx (1 - \Delta s/\epsilon)\bar{s} < \bar{s}$ in first order of $\Delta s/\epsilon$. However, when $\Delta s \approx \epsilon$, $q \approx \bar{s}/2$ is relatively insensitive to the noise $\Delta s$. In particular, when the score $s$ varies about $\Delta s \approx 0.02 = 2\% \approx \epsilon$ across the set of validations indexed by $k$, then $Q \approx 50\bar{s}$, i.e. for almost perfect $s \approx 1$ values across the board, we obtain a $Q$ close to $50\%$. Only, when $\Delta s$ significantly drops below the fixed $\epsilon = 2\%$, $Q$-scores close to $100\%$ are possible (given close to perfect $s$-scores).

In order to gather sufficient statistics to fairly compare the challenge participants, the number of linear classifiers trained on separately-sampled training and test sets was varied from $k = 1, 2, \ldots, 40$ during the development phase and $k = 1, 2, \ldots, 200$ during the final evaluation phase. While the seed for the random number generator used for the training and test set splits is kept constant for our benchmark in Section B, for the data challenge it was initialized at random. Our choice was motivated by the effort to minimize information leakage about the hidden downstream tasks to the data challenge participants. During the development phase, submissions could test the constant set of predefined downstream tasks over a three-week period and submit 10 times a day.

***Global* ranking relative to other challenge participants.** On top of a single participants $p$'s (*local*) performance score $Q_t^{(p)}$, we added a *global* ranking scheme as follows: Both local and global rankings assign rank $R_t^{(p)} = 1$ to the highest performing participant and ascending rank $R_t^{(p)}$-values for decreasing performance. Ties are broken such that all tied participants get the lower (best) rank. The algorithmic design of our approach is best illustrated in a Python code implementation like:

```python
q = {
    'team1': 13.223,
    ...,
    'teamP': -3.55677
}

def rank(q:dict, descending:bool = True) -> dict:
    sign = 1
    if descending:
        sign = -1
    return {
        p: 1 + len(
            [ s_sub for s_sub in q.values() if sign*s_sub < sign*s ]
        )
        for p, s in q.items()
```

---

[3]For an $R$-squared score (regression task), $s < 0$ penalizing good, positive values $s \in [0,1]$. In fact, negative $s$–values indicate that the downstream task prediction is worse than a model simply predicting the value of the mean label.

```
16        }
17
18  ranked_q = rank(q)
```

where the Python dictionary `q` serves as input to `rank()` to generate $R_t^{(p)}$=`ranked_q` and the boolean parameter `descending` triggers whether the highest or lowest value is deemed best. Utilizing `rank()`, the local ranking $R_t^{(p)}$ orders participant $p$ on task $t$ with the highest (best) score $Q_t^{(p)}$, `descending=True`. The second, global ranking across tasks assigns rank $R^{(p)} = 1$ to the participant with the lowest (best) weighted average rank score

$$s^{(p)} = \sum_{t=1}^{T} w_t R_t^{(p)} \quad \text{where} \quad w_t = \text{std}_p Q_t^{(p)} \Big/ \sum_{t=1}^{T} \text{std}_p Q_t^{(p)} \tag{4}$$

by setting `descending=False`. In contrast to $\text{std}_k$ over cross-validation folds $k$ in Eq. (3), here, $\text{std}_p$ runs over the number of participants $p$ of a fixed task $t$, i.e., the weight $w_t$ computes the variation of our evaluation metric $Q_t^{(p)}$ for a given task $t$ across all data challenge participants $p$. Thus, $w_t$ serves as a measure of *task competitiveness* to characterize and automatically distinguish tasks $t$.

Our design rationale of the `weighted_score` for the data challenge was as follows:

- reward participants scoring well for a given downstream task
- discount the quality score $Q$ depending on the *task competitiveness* of a downstream task, i.e., measure relative performance among challenge participants for a given downstream task.

The std-based weighting achieves this by discounting downstream tasks where all teams perform similarly, in analogy to:

> *A football match is a draw regardless if the end result is 1-1 or 8-8 — although the number of goals can have a marginal effect in a tournament.*

We assign more importance to downstream tasks where participants score high AND when they distinguish themselves from the rest. More formally speaking: For a weight $w_t = \delta_t / \sum_\tau \delta_\tau$ with $\delta_t = \text{std}_p Q_t^{(p)}$ and the commonly accepted definition of variance

$$(\text{std}_p A_p)^2 = \langle A_p^2 \rangle_p - \bar{A}^2 = \langle (A_p - \bar{A})^2 \rangle_p \tag{5}$$

where

$$\langle f(X_p) \rangle_p = \frac{1}{P+1} \sum_{p=0}^{P} f(X_p) \quad \text{and} \quad \bar{A} = \langle A_p \rangle_p \tag{6}$$

such that

$$A_p = \bar{A} \quad \Leftrightarrow \quad \text{std}_p A_p = 0 , \tag{7}$$

the case $\delta_t \to 0$ for all $t$ may generate a numerical instability. However, our two distinct competition baselines

- $p = 1$: simple data aggregation of data cubes termed *Baseline mean embeddings* in the data challenge with leaderboard mean Q-score $\langle Q_t^{(1)} \rangle_t = $`-0.786`
- $p = 0$: random embeddings termed *Baseline random embeddings* in the data challenge with leaderboard mean Q-score $\langle Q_t^{(0)} \rangle_t = $`-7.092`

prevent $\delta_t = 0$ in practice as verified by running the data challenge over a month with more than 400 submissions from over 20 teams.

From a theoretical perspective, one may want to stabilize $w_t$ by adding a *ghost task* $t = 0$ with variance $0 < \delta_0 = \epsilon \ll 1$ such that

$$\delta_0 = \sqrt{\left\langle Q_0^{(p)2} \right\rangle_p} > 0 \quad \text{setting} \quad \bar{Q}_0 = 0 \quad \text{and defining} \quad R_0^{(p)} = 0 . \tag{8}$$

Abbreviating $\sum = \sum_t \delta_t$ we distinguish the cases

- $\sum \gg \epsilon$: where $w_0 = \epsilon / \sum \ll 1$ and $w_t = \delta_t / \sum$ leaving $s^{(p)}$ of Eq. (4) intact to 0th order in $\epsilon$

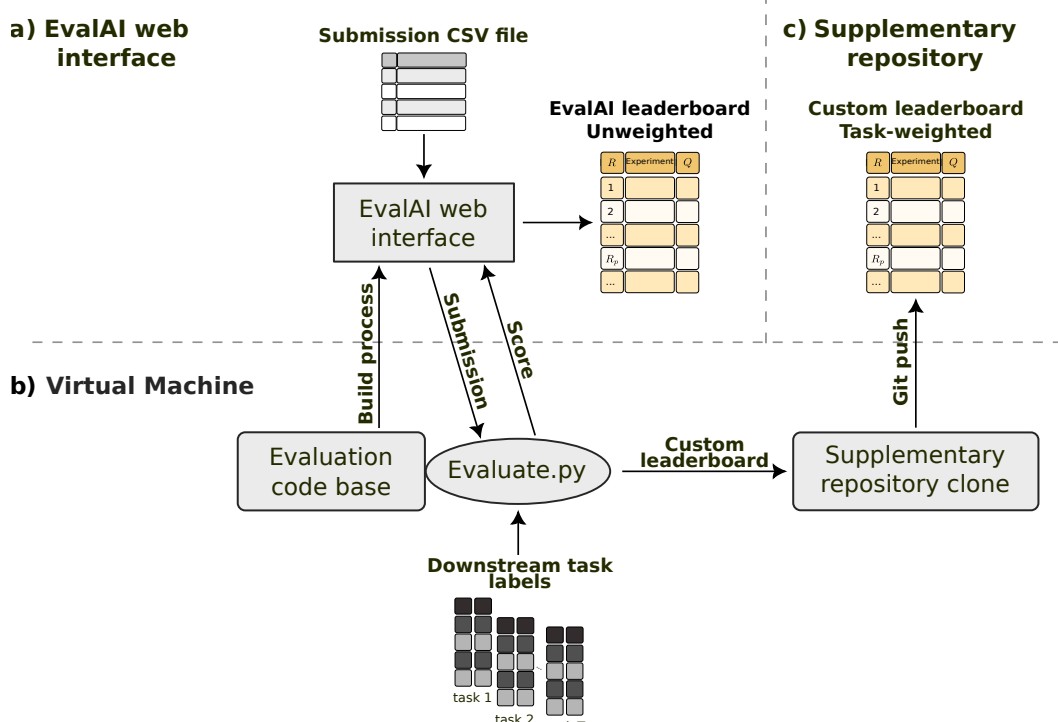

Figure 6: Components and interaction of thedata challenge. The community platform Eval.AI (a) interacts with a virtual machine hosted at a cloud service (b). The virtual machine returns the quality score of the submission to the Eval.AI leaderboard and pushes updates to the custom leaderboard (c).

- $\sum \approx \epsilon$: where $w_0 \approx \frac{1}{2}$ and $w_t \approx \frac{1}{2}\delta_t / \sum$ such that with $R_0^{(p)} = 0$ the score $s^{(p)}$ in Eq. (4) receives a discount factor $\frac{1}{2}$ which further increases for $\sum \to 0$ where $w_0 \to 1$

For the data challenge we ran our benchmark with task weighting with $1 = \sum_t w_t$. When users simply want to benchmark their neural compression methodologies on a (sub)set of downstream tasks with known complexity without competing against other teams, the unweighted averaging is the preferred mode of operation for our benchmark. In Section A.3 we report on operational insights related to task weighting as observed in the context of the data challenge. The results underline that the concept of *task competitiveness* bears further opportunities for continued research.

## A.2 PLATFORM AND INFRASTRUCTURE

The core evaluation pipeline was implemented on a virtual machine (VM) with specifications:

- Operating System/OpenStack Image: Ubuntu Jammy 22.04 LTS
- CPU: 8 vCPUs, no GPU
- RAM: 16GB
- Disk: 20GB (OS) + 200GB (data storage)

running on top of OpenStack[4] cloud environment. The communication with the Eval.AI API for fetching submission data and writing results back to the Eval.AI leaderboard was based on the Eval.AI GitHub *remote challenge evaluation* template utilizing the `requests` Python library.[5] Figure 6 illustrates the entire setup: (a) the Eval.AI web interface and a *supplementary repository* on one end, and (b) the evaluation procedure which runs on the VM at JSC, on the other end.

As evaluation method (`Evaluate.py`), our benchmark framework was incorporated into the Eval.AI remote challenge evaluation template running on the VM, cf. *Local Repository – Evaluation*

---

[4]`https://www.openstack.org`
[5]`https://docs.python-requests.org`

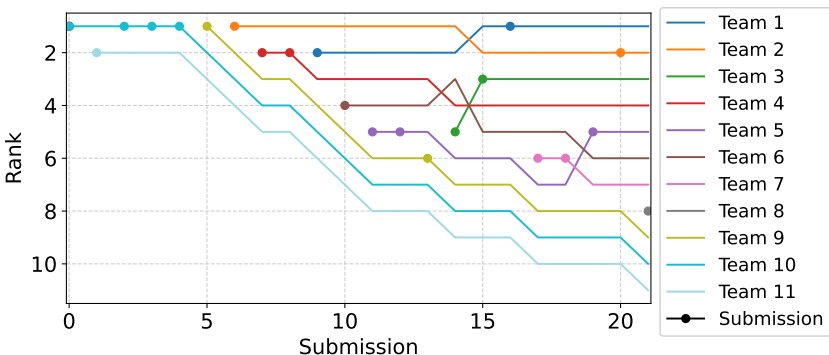

Figure 7: The evolution of participant rankings in the challenge test phase. Lines correspond to participating teams and dots to submissions by the corresponding team. Team 10 is the simple mean baseline described in Section B.3.2, and Team 11 is a randomized baseline with randomly sampled, normally distributed embeddings.

*Codebase* in Fig. 6. Updates to the Eval.AI challenge web interface got triggered by *GitHub Actions*.[6] In addition, the *Supplementary Repository* serves two purposes:

- for the challenge participants to provide instructions and code examples with options to raise issues, and
- to host a Custom Leaderboard implementing the global ranking introduced in Section A.1, not natively supported by Eval.AI

The VM runs a `cronjob` to restart `Evaluate.py` in case the application terminated. In fact, every minute our benchmark framework polls Eval.AI for new *Submissions* to score. Thereafter, the VM reports $Q$ of the evaluated submission to the *Eval.AI Leaderboard*. It also updates the global *Custom Leaderboard* in the GitHub *Supplementary Repository*.

### A.3 COMPETITION ANALYSIS

The interaction between participants and organizers through GitHub issues allowed for transparent and traceable communication. In particular, we highlight an update to the challenge that improved comparability between participants by reducing variability in case the same submission is submitted multiple times.

Other learnings from the develpment phase are:

- Normalization of the target labels across all downstream tasks may be necessary to avoid hyperparameter tuning of the linear probe.
- Before normalizing the target labels, the range of the target labels heavily affected the ability of the linear probe to learn a specific task within the given network initialization, learning rate and number of epochs.

In total, nine teams participated publicly in the final phase of the data challenge, competing over scoring top rank and highest mean $Q$ value across all tasks.

In general, the ranking and the mean $Q$ value are close to identical. However, the team achieving third place upended the order of first and second place, with the effect that the runner-up team achieved a slightly higher mean $Q$-score than the winners. This effect is driven by a change in task weights caused by the third-place team's performance. We note the dynamics around Submissions 14 and 15 as illustrated in Fig. 7 where the ranking dynamics given a sequence of *Submission*s (dots) is documented: Team 1 through 9 are competing numerically indexed by their final position in the challenge ranking. Team 10 represents the simple mean baseline case described in Section 5.3 with additional details in Section B.3.2, and Team 11 is a *randomized* baseline submitting randomly sampled, normally distributed embeddings.

A hallmark of our dynamic (global) ranking $R^{(p)}$ can be observed as follows: At submission 14, the submission of Team 3 modified the task weights such that the position of Team 4 and 5 were swapped,

---

[6]https://docs.github.com/en/actions

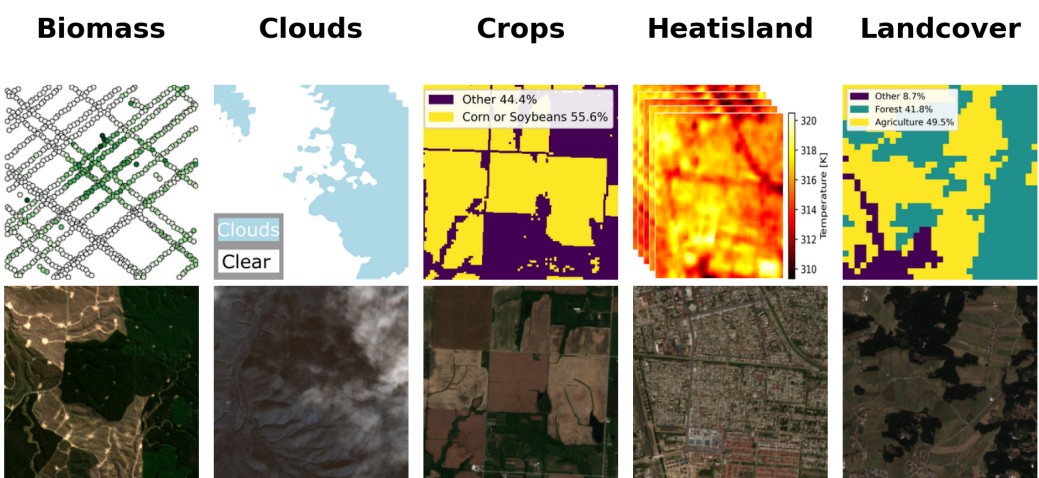

Figure 8: Visualisation of downstream-task labels (top row) and corresponding Sentinel-2 images (bottom row). The biomass labels have been sourced from Karaman et al. (2025). Although Sentinel-1 is included in every data cube, it was excluded from this visualisation. The cloud and heat island labels are based on aggregated images.

Table 2: Qualitative comparison of our benchmark, PANGAEA, and GEO-Bench

|  | ours | PANGAEA | GEO-Bench |
|---|---|---|---|
| *Domain* | General purpose compression | Geo. foundation models | Geo. foundation models |
| *Compute* | commodity hardware | AI accelerator | AI accelerator |
| *Model access* | Not required | Intermediate features | Backbone finetuning |
| *Tasks* | Classification | Classification | Classification |
|  | Regression | Regression | - |
|  | - | Segmentation (**focus**) | Segmentation |
| *Leaderboard API* | JSON | JSON | - |

even though Team 3 ranked below the two other teams. The same occurred at submission 15, where the submission of team 3 changed the task weights to the benefit of Team 1. The adaptations of the task weights were small compared to the weights of the four tasks with highest weights—on the order of a few percent of the task weights. This drastic effect on the rank of the first two positions is partly due to Team 1 and 2 being neck and neck, Team 1 winning with a weighted average rank 2.31 and Team 2 coming second with 2.44, even though Team 2 scored 15.2 mean $Q$, ahead of Team 1 on 14.9.

The proposed task weighting method as defined by Eq. (4) achieved to balance the importance of tasks. We noted that the agriculture and forest related tasks were well solved by several teams. The other tasks turned out more challenging. As expected, the (random) baselines were indicated low performers according to $R^{(p)}$. A limitation of the weighting we observed: Since the weighting of the tasks $t$ in Eq. (4) is based on the variations of the participants for that given task $t$, a participant $p$ very poorly performing by design—such as the random baseline (Team 11)—artificially inflates the task weight when all the other participants perform well. A sensible extension of our benchmark framework as discussed in Section B will depend on a careful design of downstream tasks and corresponding baselines.

## B   An Extendable Framework

Based on our insights from the data challenge, we took our approach to the next level with the intention to build a community around benchmarking neural compression. Table 2 provides a high-level comparison on how our benchmark fits into existing, popular geospatial benchmarking frameworks. In summary, our benchmark fills the following gaps:

- Quantifies the quality of small embeddings based on a variety of downstream tasks without fine-tuning of any neural network backbone.

- Provides a standalone toolkit for rapidly benchmarking any compressed embeddings beyond foundation models. In contrast to GEO-Bench and PANGAEA, our benchmark framework is readily adapted to any compression scenario given:
  - Users provide embeddings $z$ where their encoder $E$ takes care of data formats.
  - Downstream tasks are shared with our benchmark framework as simple CSV files.
- Supplies a multi-task performance metric that quantifies embedding size ($N$) vs. downstream accuracy ($Q$).

It is worth iterating that our dataset **downstream** dataset builds on data structures part of GEO-Bench—thus, serving as a potential interface regarding synergies. We are currently forming the *Earth2Vec* community around our benchmark framework where more than 30 organizations from academia, the corporate world, and governments have joined.

## B.1 BENCHMARK TASKS

Figure 8 illustrates examples of Sentinel-2 inputs alongside the corresponding labels. Table 3 lists a detailed overview of the derived 11 tasks, whereby 9 of these were used in the data challenge and are highlighted with green check marks. *Clouds* and *Nodata* were not included in the competition but are provided with the release of our benchmark framework. The task *Random* provides randomly generated labels and associated data cubes from the *Cloud* task, which introduced an additional quality assessment.

The processing pipeline for all presented datasets utilized GEE GEE to download data cubes, applying a maximum cloud coverage filter of 10%, as provided by the GEE property CLOUD_COVER, except for the clouds task where no restrictions on cloud cover were enforced. Each data cube was aligned to the center of the corresponding label and processed to a size of 264 x 264 pixels. All sample locations with less than 4 images were discarded. Whenever possible, only locations that cover all four seasons for Sentinel-1 and Sentinel-2 were chosen. In case of missing latitude and longitude, locations were randomly selected from shapefiles representing regions such as mainland Europe or areas within the US Corn Belt. show one major difference. In January 2022, ESA introduced a new baseline for Sentinel-2 data, effectively shifting all pixel intensities by 1000 units upward. The dataset presented in this work follows the format of GEE, i.e. removing this translation such that the minimum value for Sentinel-2 pixels are 0 both before and after the change by ESA. To allow for seamless integration between the two datasets, the dataloaders provided in our benchmark includes a setting that toggles a shift by 1000, aligning the distributions of the two datasets.

The *Heat-island* task required additional pre-processing, as Landsat-8 band 10 (B10) was utilized for label generation. This dataset considers only cities with populations exceeding 20,000 and a latitude between $8°$ and $70°$ north. The labels are based on all available Landsat-8 observations from June to the end of August for the years 2021 to 2024 inclusive. In addition, to reduce the impact of remaining clouds, any pixel with a combined brightness (red channel + blue channel + green channel) exceeding 30% of the maximum possible value or with a B10 temperature lower than 273 K are removed. Images with more than 10% removed pixels were dropped. The northernmost locations were verified to have average summer temperatures above freezing. For each location, the remaining images are flattened and concatenated over time, and then the mean and standard deviation are calculated from all pixels. The task is to estimate these spatio-temporal statistics.

**Public vs. Secret Downstream Tasks.** We released the hidden downstream tasks (cf. Table 3 with green check mark) after the conclusion of the workshop to make publicly accessible the standalone downstream dataset for reasons of transparency, and to be used and contributed to by the neural compression community. As common with public benchmarks designed for standalone usage, we assume that benchmark users would not jeopardize the developments of their own compressor $E$ by willingfully exploiting knowledge of the downstream tasks they test on. Removing and adding (mix-and-match) downstream tasks for a new competition avoids overfitting of a state-of-the-art compressor $E$. The process is as straightforward as uploading such data $x$ to a file sharing service, i.e.,

1. *public*: Each data point $x_i$ just needs a unique (identified by hash $i$) name (cf. e.g., directory `data/` of downstream dataset) to upload corresponding ...
2. *secret till conclusion of competition*: ...label CSV files (cf. e.g., directory `labels/` of downstream dataset, `id` column of CSV file) ...

...for the benchmark engine to perform its downstream task evaluations.

Table 3: Descriptions of the downstream tasks provided by the initial release of our benchmark. The tasks used in thedata challenge are indicated with green check marks in the "Challenge" column. The task names correspond to the identifiers as used in the corresponding dataset released.

| Task | file | Challenge | Description |
|------|------|-----------|-------------|
| **Biomass** | biomass_mean__regr | ✓ | Regression tasks: Biomass density (Mg/ha) mean and standard deviation estimated for pixel-level labels derived from GEDI. |
| | biomass_std__regr | ✓ | |
| **Crops** | crops__regr | ✓ | Regression task: Combined fraction of Soybean and Corn in the label image. |
| **Landcover** | landcover_agriculture__regr | ✓ | Regression task: Percentage of agriculture pixels in the Corine Land cover image. |
| | landcover_forest__regr | ✓ | Regression task: Percentage of forest pixels in the Corine Land cover image. |
| **Clouds** | clouds_reg__regr | ✗ | Regression task: Average cloud cover fraction across four seasons in one year. |
| **Heatisland** | heatisland_mean__regr | ✓ | Regression tasks: Summer surface temperature mean and standard deviation in Kelvin. |
| | heatisland_std__regr | ✓ | |
| **Nodata** | nodata__regr | ✗ | Regression task: Fraction of pixels with value zero in a Sentinel-2 image (based on all 13,260 available samples). |
| **Random** | random_reg__regr | ✓ | Regression task: Random task with a majority of zero labels. The data cubes are the same as for Clouds. |
| | random_cls__cls | ✓ | Classification task: Random binary classification for a majority of zero labels. The data cubes are the same as for Clouds. |

## B.2 STANDALONE PYTHON IMPLEMENTATION

In order for a clean separation of code from the open-source platform Eval.AI, we developed a minimal viable standalone Python code base to serve as plug-and-play for any larger ecosystem integrating of the core framework. In fact, as Fig. 6 demonstrates, the scoring for the Eval.AI leaderboard is entirely taken care of by our benchmark. Correspondingly, our framework commits an additional, customized leaderboard that *globally* depends on all submissions to a dedicated GitHub repository.

The benchmark core `evaluation.py` functionality separately fetches

- the user's embeddings (submission), `/path/to/submission_file.csv`, and
- the downstream task annotation data (labels), `/path/to/annotation_directory/`

as ASCII-formatted `CSV` files given predefined local paths and directories as simple interface entirely independent of Eval.AI. Given any ranking procedure implemented, the resulting leaderboard is saved as human-readable `JSON` file in a corresponding `/path/to/results_directory/`. For downstream (binary) classification tasks, the confusion matrix and related scores such as precision, recall, F1, and overall accuracy are calculated along with the ROC-AUC-score (area under Receiver-Operator-Characteristic graph). For regression, the R-squared, mean squared, and mean absolute errors are computed.

To serve as seed towards an open-source and open science community, we designed the standalone Python implementation of our benchmark modular for easy extension. Depending on compute resources, we encourage future contributions to add novel probing models, cross validation schemes, and performance scores (cf. Eq. (3)) beyond the current. As a bonus, our standalone implementation

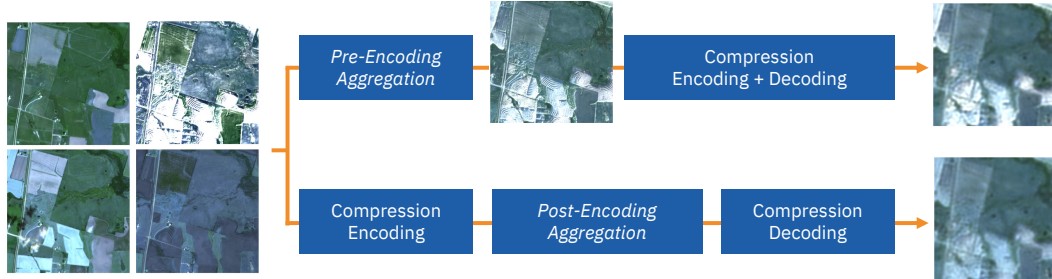

4-Timestep Input Data

Figure 9: Illustration of pre-encoding vs. post-encoding aggregation. In post-encoding, each seasonal image is encoded separately before combining embeddings, which mitigates outlier effects (e.g., snow) but increases runtime fourfold.

allows to store plots of loss curves, linear correlation of regression tasks, and a confusion matrix for classification on disk.

Running our benchmark standalone on the command line reduces to something as simple as:

```
python main.py \
    --annotation_path /path/to/annotation_directory/ \
    --submission_file /path/to/submission_file.csv \
    --output_dir /path/to/results_directory/ \
    --config /path/to/config.yaml \
    --method_name 'your-method-name' \
    --phase 'phase-name'
```

where `your-method-name` and `phase-name` are free strings to define an output (sub-)directory

```
    /path/to/results_directory/phase-name/your-method-name_YYYYMMDD_HHmmSS
```

with `YYYYMMDD` a date of year `YYYY` and zero-padded numerical month `MM` and day `DD`. `HHmmSS` indicates a time of the day in hours `HH`, minutes `mm`, and seconds `SS`, accordingly. A `YAML` file `/path/to/config.yaml` specifies details of the evaluation such as:

```
embedding_dim: 1024          # number of embedding dimensions
batch_size: 64               # batch size for (linear) probing
epochs: 20                   # number of epochs to optimize the (linear) probe for
learning_rate: 0.001         # learning rate to optimize with
k_folds: 40                  # number of cross-validations to generate statistics over
standardize_embeddings: true # standardize embeddings by their global mean and std
normalize_labels: true       # normalize in range [0,1]
task_filter: false           # all in /path/to/annotation_directory/ per default
                             # example: ["biomass_mean", "biomass_std"]
# etc.
```

## B.3 DETAILS ON BASELINE METHODS

This appendix expands on the general evaluations introduced in Section 5.3 by providing methodological details and extended results.

Unlike the challenge setting, which separated development and evaluation splits, all results in Section 5.3 and this appendix are computed on the full downstream datasets. Unless noted otherwise, we follow the evaluation protocol introduced in the main text, and use $E = 20$ training epochs, $k = 50$ train–test splits, and a learning rate of $10^{-3}$. We report raw $R^2$ values, clipping negative scores to $[0, 1]$ only for visualization.

We begin by comparing the temporal aggregation methods in Section B.3.1 which motivate the use of post-encoding aggregation for all subsequent analyses. Thereafter, we report additional baseline methods and results for the 1,024-dimensional embedding setup Section B.3.2. In Section B.3.3 we extend the ablations introduced in Section 5.3 by providing per-task results for varying embedding dimensions and decoder choices beyond linear probing.

Table 4: Comparison of temporal aggregation methods. We show average $\bar{R}_p^2$ and $\bar{R}_P^2$ scores across all downstream tasks (Sections B.1 and 4) for pre-encoding vs. post-encoding aggregation, respectively. We also provide the overall improvement $\boldsymbol{\Delta R^2 = \bar{R}_P^2 - \bar{R}_p^2}$ and the best gain per task, **Best $\boldsymbol{\Delta R^2}$**.

| Method | Pre-Encoding $\bar{R}_p^2$ | Post-Encoding $\bar{R}_P^2$ | $\boldsymbol{\Delta R^2}$ | Best $\boldsymbol{\Delta R^2}$ (task) |
|---|---|---|---|---|
| Averaging Baseline | -0.522 | -0.385 | +0.137 | +0.270 (clouds) |
| Factorized Prior | 0.238 | 0.233 | -0.005 | +0.044 (clouds) |
| DINO ResNet (FM) | 0.289 | 0.397 | +0.108 | +0.318 (clouds) |
| DINO ViT (FM) | 0.382 | 0.470 | +0.088 | +0.423 (clouds) |
| MAE ViT (FM) | 0.481 | 0.537 | +0.056 | +0.272 (clouds) |
| TerraMind (FM) | 0.600 | 0.659 | +0.059 | +0.297 (clouds) |

### B.3.1 TEMPORAL AGGREGATION ANALYSIS

Here, we study image-based encoding methods for our baseline evaluations. As each input sample contains four seasonal Sentinel-1/2 snapshots, we handle temporal sequence data by reducing the four snapshots into a single embedding using two aggregation strategies:

- **Pre-encoding:** seasonal snapshots are averaged before encoding.
- **Post-encoding:** each snapshot is encoded separately and the resulting embeddings are averaged. As shown in Fig. 9, this approach better handles seasonal outliers (e.g., snow) at the cost of additional computation.

Across all methods considered in the main paper (Figure 4), post-encoding aggregation provides consistent $R^2$ improvements for most methods and tasks as summarized in Table 4. The largest gains occur for cloud-fraction prediction, with increases of up to +0.42 for ViT-based encoders and +0.30 for TerraMind. Semantic tasks such as land-cover show smaller but consistent improvements (+0.011 to +0.016). In summary, post-encoding yields performance gains, in particular for temporally sensitive tasks such as determination of cloud fractions. These results motivate the use of post-encoding aggregation for all the analysis below.

### B.3.2 EXTENDED FIXED-SIZE EMBEDDING EVALUATIONS

In the following we extend the results of Section 5.3 with additional GeoFMs and an analysis of predicting with CLS tokens instead of averaged patch tokens. All experiments in this section are carried out with a fixed embedding dimension of 1,024.

**Averaging Baseline.** As a simple informative reference, we construct a *Mean baseline* by strongly downsampling and averaging the data cubes. First, we reduce the spatial resolution of each of the 27 channels from 264x264 pixels to 8x8 by bi-linear interpolation. Next, we exploit correlation as visualized by Fig. 10 reducing the number of channels from 27 down to four. We average the channels B1 through B9 of both S2L1C and S2L2A and we do similar for channels B11 and B12, respectively. Channel B10 of S2L1C is kept separate since no corresponding band exists in S2L2A. Seasonal snapshots are kept separate, yielding 8×8×4×4 values, flattened to $N = 1024$. This baseline indicates how much task-relevant information survives coarse spatial and spectral aggregation.

**Neural rate–distortion compressors.** We adopt the Factorized Prior autoencoder of Ballé et al. (2016) for Sentinel-2 L1C imagery. Models use 256 intermediate channels and 128 latent channels, trained with loss

$$\mathcal{L} = R + \lambda D \tag{9}$$

where $D$ is MSE distortion and $\lambda \in \{0.025, 0.1, 0.5\}$ and $R$ represents the entropy-coding term for bit-stream compression. These compressors match or exceed JPEG2000 PSNR at roughly half the bitrate (Fig. 11). At inference, we pool latents to $4 \times 4$, flatten to 2048, and average adjacent channels to yield 1024-dim embeddings.

**Self-supervised foundation models (FMs).** For a comprehensive overview, we evaluate a broad set of pretrained EO FMs—including those used in the main paper. We include Clay-V1 ViT-B (Clay Foundation Model, 2025), Prithvi-EO-V2-300M (Szwarcman et al., 2025), TerraMind-V1-B (Jakubik et al., 2025), DOFA-ViT-L (Xiong et al., 2024), Satlas Multispectral ResNet-50 (Bastani et al., 2023), SSL4EO-pretrained ResNets (CNN, MoCo) and ViT-B/16 backbones (DINO, MoCo, MAE) (Wang et al., 2023b).

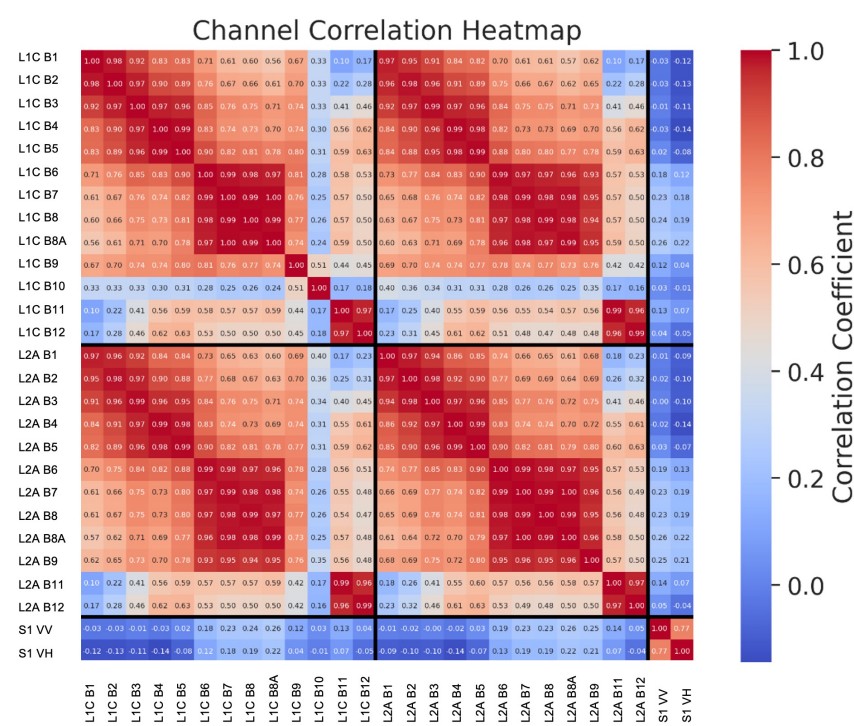

Figure 10: Pearson correlation coefficient matrix of the 27 data cube channels. In here, we abbreviate the channels of the Sentinel-2 L1C and L2A products as L1C and L2A, respectively appending the channel name (B1, B2, . . . ).

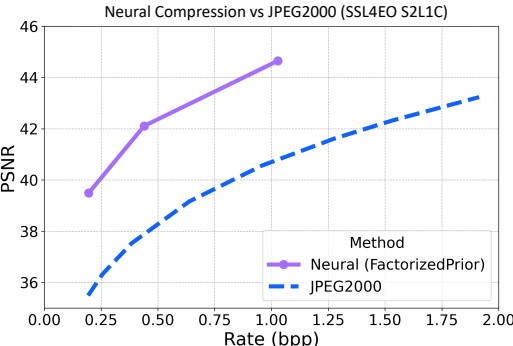

Figure 11: Rate–distortion performance of the Factorized Prior neural compressor, demonstrating superior compression quality over the JPEG 2000 baseline.

As introduced in Section 5.3, we apply a consistent embedding aggregation pipeline across all FMs. As motivated by Section B.3.1, we utilize temporal post-encoding aggregation throughout our experiments. CNN outputs are reduced via global average pooling. If the pooled feature dimension exceeds 1,024, we apply pairwise channel means, i.e., adjacent feature channels are averaged in pairs to halve the dimensionality while preserving coarse channel structure. For ViT encoders, we average the spatial patch tokens (excluding the CLS token, if present) to form a single embedding. Additionally, we also evaluate CLS-token embeddings for Prithvi and Clay. All aggregated embeddings are padded to the target 1,024-dimensional space, if any. Table 5 provides a consolidated view of linear-probe performance across all tested embedding methods. Overall, we observe the following trends:

- **FM Embeddings.** Multimodal models, most prominently TerraMind, consistently outperforms all other embeddings, achieving the highest $R^2$ across most tasks. We highlight that

Table 5: Full per-task $R^2$ scores for tested embedding methods, including average performance (Avg.) across all tasks. For ViT's, experiments utilizing the CLS token for prediction is indicated by (CLS), otherwise the average patch token is used. Methods are sorted in ascending order by Avg. For each task, the best-performing method is highlighted in **bold**, and the second-best method is underlined.

| Method | Biomass Mean | Biomass Std | Crops | Clouds | LC Agri | LC Forest | HI Mean | HI Std | Avg. |
|---|---|---|---|---|---|---|---|---|---|
| Averaging Baseline | -0.552 | -0.426 | -0.061 | -2.293 | 0.126 | 0.216 | -0.962 | -1.157 | -0.514 |
| FP ($\lambda = 0.5$) | -0.036 | -0.071 | 0.325 | 0.200 | 0.478 | 0.343 | 0.033 | -0.628 | 0.080 |
| FP ($\lambda = 0.1$) | 0.129 | 0.078 | 0.357 | 0.266 | 0.464 | 0.333 | 0.219 | -0.270 | 0.197 |
| FP ($\lambda = 0.025$) | 0.195 | 0.140 | 0.338 | 0.288 | 0.449 | 0.256 | 0.315 | -0.113 | 0.233 |
| S12-DINO ResNet50 | 0.117 | 0.088 | 0.826 | 0.147 | 0.879 | 0.865 | 0.483 | -0.221 | 0.286 |
| S12-MoCo ResNet50 | -0.048 | -0.030 | 0.780 | 0.216 | 0.864 | 0.844 | 0.345 | -0.334 | 0.286 |
| Satlas MS ResNet50 | 0.310 | 0.248 | 0.490 | 0.340 | 0.658 | 0.641 | 0.485 | 0.092 | 0.408 |
| Clay-V1 ViT-B (CLS) | 0.199 | 0.169 | 0.748 | 0.482 | 0.811 | 0.803 | 0.430 | -0.013 | 0.454 |
| S12-DINO ViT-S | 0.184 | 0.183 | 0.842 | 0.484 | 0.863 | 0.851 | 0.461 | -0.109 | 0.470 |
| S12-MoCo ViT-S | 0.338 | 0.259 | 0.751 | 0.409 | 0.825 | 0.814 | 0.506 | 0.133 | 0.504 |
| DOFA-ViT-L | 0.373 | 0.269 | 0.587 | 0.518 | 0.777 | 0.773 | 0.560 | 0.207 | 0.508 |
| S12-MAE ViT-S | 0.380 | 0.280 | 0.627 | 0.659 | 0.792 | 0.797 | 0.585 | 0.181 | 0.537 |
| Prithvi-EO-V2-300M (CLS) | 0.426 | 0.329 | 0.655 | 0.531 | 0.764 | 0.782 | 0.603 | 0.216 | 0.538 |
| Prithvi-EO-V2-300M | 0.424 | 0.324 | 0.671 | 0.588 | 0.781 | 0.792 | 0.608 | 0.203 | 0.549 |
| Clay-V1 ViT-B | 0.466 | 0.332 | 0.759 | 0.660 | 0.825 | 0.825 | 0.606 | **0.229** | 0.588 |
| TerraMind-V1-B | **0.528** | **0.390** | **0.879** | **0.731** | **0.918** | **0.908** | **0.691** | 0.226 | **0.659** |

jointly modelling Sentinel-1/2 can provide task benefits under strong spatial and spectral aggregation. DOFA, while scoring below TerraMind, still achieves consistent performance across all task. EO-specific, single-modal ViTs such as Prithvi and Clay follow below TerraMind and offer strong results across semantic and geophysical tasks. SSL4EO-pretrained models exhibit complementary strengths, that is: CNN variants perform strongly on landcover tasks, but fall behind ViTs on geophysical regressions. Notably, S12-DINO ResNet scores the second highest on both the land-cover agriculture and forest tasks. MAE ViT achieves balanced and high performance across tasks, while contrastive DINO and MoCo excel on semantic land-cover tasks. However, DINO/MoCO are less competitive on geophysical regression.

- **CLS Token Embeddings.** The comparison between patch-averaged and CLS-token embeddings for Prithvi and Clay demonstrates that mean patch-token averaging is the more robust strategy: for Prithvi, CLS performance is slightly lower but remains close, whereas for Clay the CLS token underperforms patch averaging. This supports our choice of patch averaging for the ViT backbones.

- **Neural compressors.** We evaluate three rate–distortion settings, which control the weight of the entropy loss term during training, cf. Eq. (9). We observe that the model with the strongest emphasis on compression ($\lambda = 0.025$) performs best overall, while lower compression focus ($\lambda = 0.5$) degrades performance. However, all variants score behind FM embeddings and struggle with the strong spatial–temporal aggregation and linear-probing setup.

- **Intermediate Layers.** To probe how different layers capture task-relevant features, we extracted intermediate embeddings from the contrastive ViTs. As shown in Fig. 12, low-level regression tasks (biomass mean/std, heat-island mean/std, cloud fraction) peak at shallow layers (layer 1), whereas higher-order semantic tasks (crop fraction, landcover proportion) reach their best performance at deeper layers.

### B.3.3 NON-LINEAR PROBING & EMBEDDING SIZE ABLATIONS

**Linear vs. Non-Linear Probing.** As part of our evaluation design, we explored the impact of decoder complexity on downstream task performance. While linear probing is the default protocol, we deliberately investigated non-linear alternatives to assess whether additional decoder capacity meaningfully improves results.

Figures 5b and 13 compare linear probes with one- and two-hidden-layer MLP decoders. We observe three consistent findings:

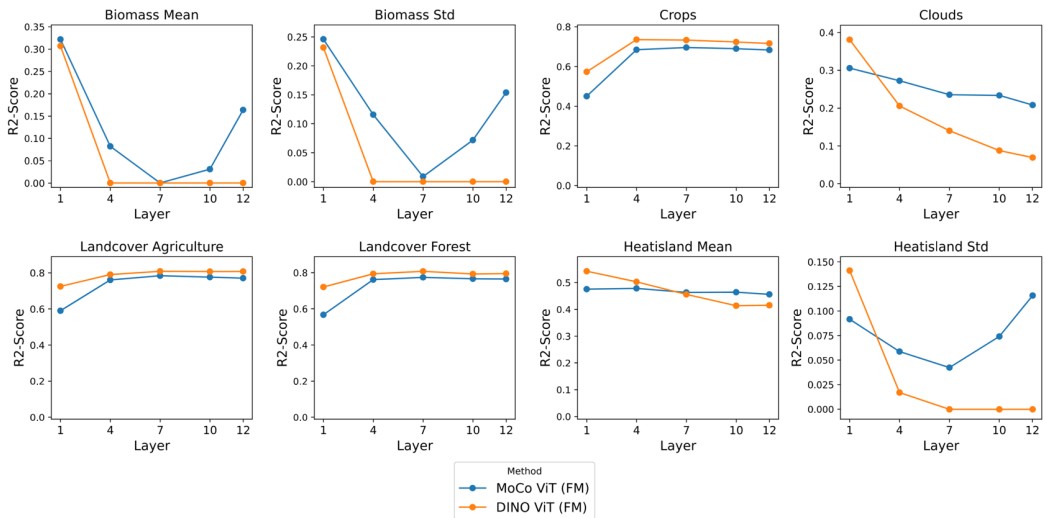

Figure 12: Layer-wise $R^2$ performance of ViT-based foundation models across downstream tasks. Shallow layers capture low-level signals useful for regression tasks (e.g., biomass, heat-islands, clouds), while deeper layers improve performance on semantic classification tasks (e.g., crops, landcover).

- **Stable rankings.** The relative ranking of embedding methods remains nearly the same across probe types, indicating that differences between methods are not an artifact of probe capacity.
- **Marginal gains for strong embeddings.** Top-performing embeddings (e.g., TerraMind, MAE) improve by less than 0.06 $R^2$ on average when switching to non-linear probes, demonstrating that these embeddings are already highly linearly expressive.
- **High computational overhead.** Increasing decoder depth leads to $\sim170\times$ and $\sim464\times$ more parameters for one and two hidden layers, respectively, with only small performance gains.

Interestingly, weaker embeddings benefit disproportionately from non-linear probes, suggesting that added decoder complexity can compensate for lower-quality representations. However, this comes at substantial computational cost.

Taken together, these results highlight that linear probing is not only efficient but also a discriminative evaluation strategy: it faithfully reflects the intrinsic quality of embeddings while enabling scalable benchmarking. Non-linear decoders may be useful for future extensions to more complex tasks (e.g., pixel-wise segmentation), but for the image-level tasks studied here, linear probing provides a robust and interpretable measure of embedding quality.

**Embedding Size Ablations.** Figures 14 and 15 show ablation results on embedding dimensionality for ViT-based and CNN-based models, respectively. For CNN backbones, performance generally peaks in the range of 128–1024 dimensions, with larger or smaller embeddings leading to consistent performance drops. ViT-based embeddings, by contrast, are most effective at their natural patch-token dimension, and reductions tend to degrade task performance. Notably, the benefit of larger embeddings is limited: increases beyond 1024 dimensions yield negligible accuracy improvements while substantially raising computational demands and probe parameter counts. These results justify the use of 1024-dimensional embeddings as a balanced default in the run data challenge, while also illustrating our benchmark's flexibility for exploring embedding-size vs. utility trade-offs in future studies.

### B.4 COMPUTE RESOURCES & TUNING PARAMETERS

At the data challenge, our benchmark completed evaluations for a single submission within about 10 minutes for the embedding dimension of $N = 1024$, the number of epochs per fold are $E = 20$, the number of training- and test set splits had been set to $k = 40$ in the development phase and $k = 200$

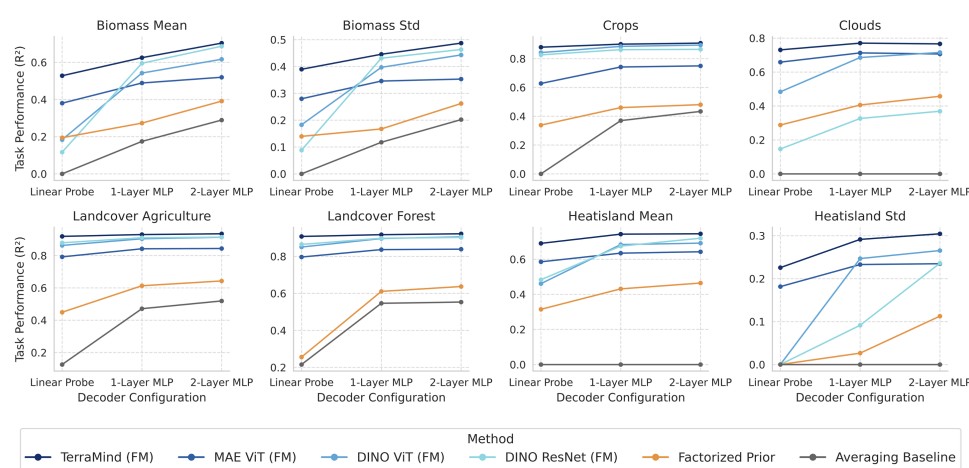

Figure 13: Per-task results for linear vs. non-linear probes. Non-linear decoders benefit weaker embeddings but have little effect on top-performing methods.

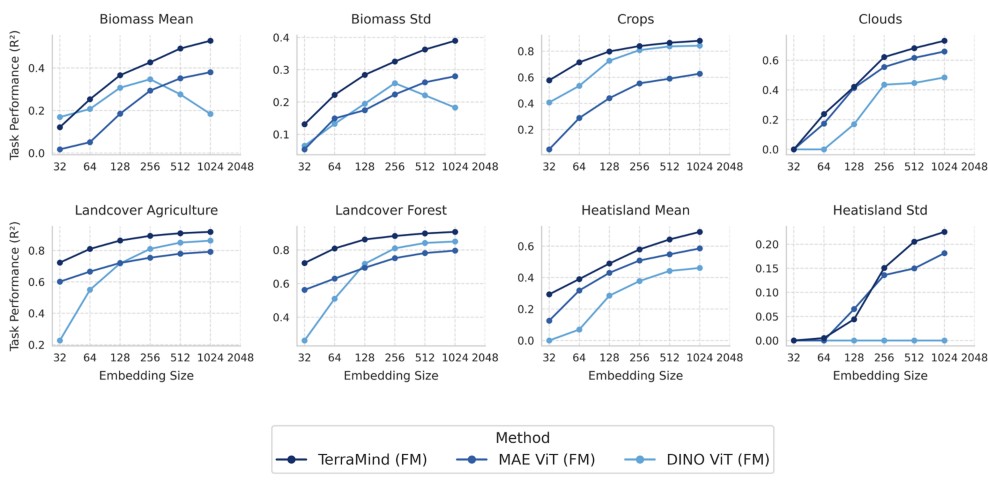

Figure 14: Embedding size ablation for ViT-based models. Performance peaks at the native patch embedding size and drops with reduced dimensions.

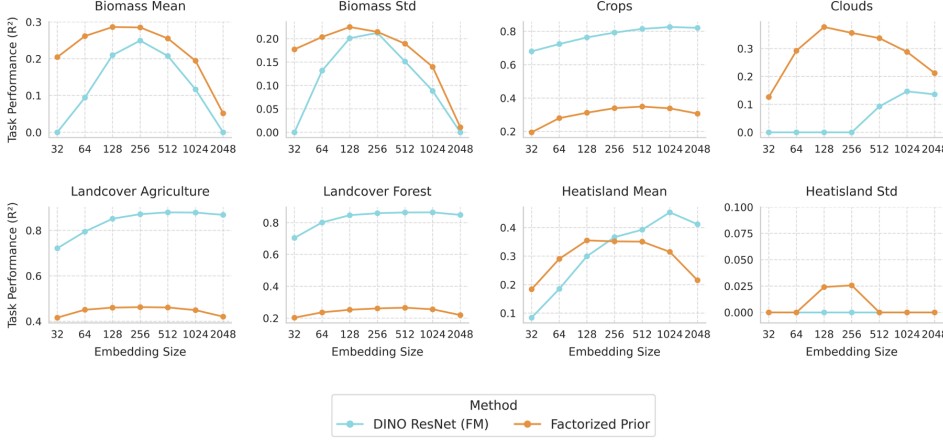

Figure 15: Embedding size ablation for CNN-based models. Optimal performance occurs between 128–1024 dimensions, with degradation outside this range.

Table 6: Empirical runtime (in seconds) for different tasks under varying embedding size ($N$), number of epochs ($E$), and number of CV folds ($k$) on single-CPU commodity hardware. Vertical lines separate configurations with different embedding sizes.

| Task (# samples) | $N = 1024$ | | | | | $N = 512$ | | $N = 2048$ | |
| --- | --- | --- | --- | --- | --- | --- | --- | --- | --- |
| | $E = 10$ $k = 40$ | $E = 20$ $k = 20$ | $E = 20$ $k = 40$ | $E = 40$ $k = 40$ | $E = 20$ $k = 80$ | $E = 20$ $k = 20$ | $E = 20$ $k = 40$ | $E = 20$ $k = 20$ | $E = 20$ $k = 40$ |
| Biomass (2415) | 4.23 | 4.24 | 7.98 | 16.01 | 16.54 | 3.87 | 7.54 | 4.25 | 8.32 |
| Crops (3355) | 5.60 | 5.56 | 11.26 | 22.15 | 22.00 | 5.09 | 10.35 | 5.94 | 12.34 |
| Clouds (1140) | 2.04 | 2.03 | 3.94 | 7.76 | 7.59 | 1.91 | 3.69 | 2.13 | 4.12 |
| Landcover Agriculture (4691) | 7.70 | 8.06 | 15.75 | 31.03 | 30.75 | 7.08 | 14.26 | 8.07 | 16.40 |
| Landcover Forest (4691) | 7.86 | 7.81 | 15.48 | 31.06 | 30.47 | 7.08 | 14.37 | 8.06 | 16.64 |
| Heatisland (1659) | 2.79 | 2.80 | 5.53 | 10.84 | 10.83 | 2.66 | 5.25 | 2.89 | 5.85 |
| No-data (13260) | 22.45 | 22.45 | 44.30 | 88.68 | 88.18 | 20.46 | 42.03 | 23.18 | 46.60 |

in the evaluation phase. The evaluation script ran across a diverse set of eight downstream tasks for real-world geospatial applications. As alluded in Section B.2, users can flexibly adjust evaluation parameters in order to tune the runtime of the standalone implementation:

- Embedding dimension ($N$, `embedding_dim`)
- Number of epochs per CV fold ($E$, `epochs`)
- Number of CV folds ($k$, `k_folds`)
- Choice of tasks included (`task_filter`)

Empirical runtime measurements confirm an approximately linear scaling w.r.t. both, the number of epochs $E$ and the number of cross-validation folds $k$. A similar scaling behavior was numerically verified for the dataset size (# samples) for fixed downstream task. Runtimes are further influenced——though to lesser extent——by the embedding dimensionality $N$. For example, increasing the embedding size from 512 to 1024 dimensions results in a runtime increase of approximately 5% to 10% across tasks. For $N = 1024$ to $N = 2048$ dimensions implies an additional increase of about 5% to 15%—depending on the task dataset size. Such a sub-linear scaling may be attributed to computation overheads and system-level inefficiencies. Those may reduce the relative computational costs when increasing the embedding dimensionality $N$. Table 6 lists a collection of recorded execution times (in seconds) for various parameter configurations per downstream task. All runtimes were measured on a single commodity ARM64 CPU with 16 cores (4.06 GHz) and 64 GB of RAM.

### B.5 LICENSES FOR DATA & SOFTWARE

Our benchmark builds on open-source software and is released under the Apache 2.0 license. All package dependencies are listed in the `requirements.txt` file, and those are licensed under widely-accepted open-source terms[7], including BSD, MIT, PSFL, The Unlicense[8], MPL-2.0[9], and Apache. These permissive licenses allow academic research and commercial use, making them fully compatible with the chosen Apache 2.0 license. Table 7 lists all data currently included in our benchmark along with their origin. Google Earth Engine (GEE) (GEE) was utilized as the primary platform for downloading downstream task data as introduced in Section B.1. Future data and code contributions to our benchmark are required to be licensed under CC-BY 4.0 and Apache 2.0, respectively.

---

[7] https://opensource.org/licenses

[8] code-equivalent to CC0 data licenses

[9] weak copy-left that allows for integration with non-copyleft licenses

Table 7: List of licenses related to datasets currently included in our benchmark. All of these, except *Clouds*, are available in GEE. Table 1 lists years of target labels, ranging from 2018 through 2024.

| Dataset | Origin of Data | License |
|---------|----------------|---------|
| Sentinel-1 & -2 | ESA / Copernicus | CC BY-SA 3.0 IGO |
| Landsat-8 | USGS (Observation and Center, 2020) | Public Domain |
| CDL | USDA NASS Cropland Data Layers (Boryan et al., 2011) | Public Domain |
| CORINE | European Environment Agency (EEA), European Union Copernicus Land Monitoring Service (European Environment Agency (EEA), 2018) | Full, Open, and Free Access |
| CloudSen12+ | CloudSEN12 project (Aybar et al., 2024) | CC0 1.0 |
| GEDI | NASA (Dubayah et al., 2022) | Public Domain |

We note that the current implementation of our benchmark lists CUDA packages covered by a proprietary NVIDIA license[10]. However, we do neither bundle nor redistributes corresponding binaries. Users and contributors to our benchmark that share related docker containers need to explicitly attribute NVIDIA's license. Fortunately, and as alluded in Table 6 and Section A.2, our benchmark runs swiftly in a VM with commodity hardware specifications on CPU compute, only. Accordingly, the standalone implementation introduced in Section B.2 can be started with (Bash) environment variable `CUDA_VISIBLE_DEVICES="` to avoid usage of GPU resources.

---

[10]`https://docs.nvidia.com/cuda/eula/index.html`

