# OpenReview forum: "A Novel Benchmark Framework for Neural Embeddings in Earth Observation"
_ICLR.cc/2026/Conference — Submitted to ICLR 2026_

### Official Review · Reviewer_vNTR · 2025-10-26

**Soundness:** 4
**Presentation:** 3
**Contribution:** 4
**Rating:** 8
**Confidence:** 5

**Summary:**

SUMMARY: This paper introduces a new benchmark Benchmarks specifically aimed at EO data embeddings. The benchmark is simple by design and includes both regression and classification tasks that have to be solved given an embedding input. The authors outline how they test their benchmark in a challenge, report results and learning and outline efforts for expanding the benchmarking and community building efforts; details here are omitted, presumably to maintain the anonymity of the submission. The authors nonetheless test a few baseline methods on their benchmarks and discuss outcomes.

**Strengths:**

This is a timely and relevant paper. I especially appreciate the following aspects:

- Benchmarks specifically aimed at EO data embeddings are desperately needed. That basically every new geo embedding model / paper evaluates on different tasks (Alpha Earth most recently) emphasizes this. Geo embedding benchmarks are also not directly comparable to vision benchmarks in the remote sensing domain, which often focus on fine tuning and specialized architectures, rather than linear probing.

- The paper is very simple - and that is a strength. It is easy to follow and understand. Same goes for the benchmark itself. Very simple, no big compute needed; extremely accessible.

- The projects focus on community building and the clear outlines of expanding this work are great to see!

**Weaknesses:**

- Given that it is such a simple benchmark, it would have been cool to see comparisons to more GeoFM based embeddings, e.g. Clay, Prithvi or AEF.

- Lack of implicit neural representations as a way to obtain geo embeddings; they are initialized by location, not image, but could still be tested using this framework (I appreciate that this might be beyond the scope of the paper) and should definitely be present in the related work section.

- Spatial coverage is, as unfortunately with pretty much all EO benchmarks, favoring Western countries.

- I understand that given that this is an anonymous submission not more details can be revealed about the challenge, but what are the authors plans for changing this in a potential camera-ready version? How would the challenge outcomes be presented then?

- Why is it necessary to constrain embeddings to exactly 1024 dimensions? Wouldn't relaxing this constraint allow for a more direct comparison of different methods?

**Questions:**

Overall this is a super relevant paper. The review is quite short, but that's because I am mostly happy with this paper. I'd ask the authors to consider my questions and concerns in the "weaknesses" section, but overall, this is already a clear accept to me. We need more benchmarks on this specific topic and conducted in such a community and access-centric way at ICLR.

---

> ### Author Response · Authors · 2025-11-19
>
> We thank the reviewer for the positive feedback.  Revised contents we mark green in the manuscript. Here is our response on questions and aspects touched:
>
> 1. __Benchmarking other GeoFMs__
>    We agree that further comparisons would be valuable. By the end of this year, we plan to add several GeoFMs (e.g., Prithvi, Clay), subject to licensing and feasible preprocessing (e.g., downsampling requirements). Corresponding results, we are going to include in the revised, camera-ready version of our manuscript. Irrelevant to this paper, but FYI: We are already collaborating with the data challenge winners on an extended study that will include additional GeoFM-based embeddings.
>
> 2. __Implicit Neural Representation (INRs)__
>    INR-based embeddings are an important and active direction of geospatial data representaions. We agree they should be acknowledged in Sect. 2, *Related Work*. We added representative references, including SatCLIP (Klemmer et al. 2025), INR-based hyperspectral compression (Zhang et al. 2024; Rezasoltani & Qureshi 2024), remote sensing INR compression (Li et al. 2023), and foundational INR/NeRF compression works (Strümpler et al. 2022; Dupont et al. 2021; Sitzmann et al. 2020).
>
> 3. __Geospatial Bias__
>    We acknowledge the Western-centric coverage---an issue common to many EO benchmarks. Our dataset is intended as a first step, not a final standard. Because the framework accepts any dataset with the expected embedding/label structure, we hope it is straightforward for the global community to contribute additional regions and move towards more geographically balanced evaluations.
>
> 4. __Challenge Disclosure__
>    The camera-ready submission will include references to the challenge webpage, leaderboard, code repositories for reproducing open-sourced solutions, and guidance on contributing new datasets or evaluation modules. We are also organizing monthly community calls with invited talks on EO embeddings with perspective to host follow-up community challenges.
>
> 5. __Rationale for the 1024-dimensional constraint__
>    The limit to the 1024-dimensional embedding was imposed only for the purpose of the data challenge, to ensure a fair comparison under a fixed compression ratio. Without such a constraint, participants could trivially submit a near-linearized version of the raw datacube. The chosen value represents a strong compression factor (of 7k), remains below the available label counts per task, and is sufficiently large to allow a diverse set of methods.
>    Importantly, the benchmark framework itself places no restriction on embedding size. As shown in Fig. 5 and Tab. 5, other dimensions are already supported, and future framework extensions---particularly towards neural compression---may explicitly incorporate embedding dimensionality or compression rate as an additional evaluation metric.

---

### Official Review · Reviewer_uuFe · 2025-11-01

**Soundness:** 2
**Presentation:** 3
**Contribution:** 2
**Rating:** 6
**Confidence:** 4

**Summary:**

The authors propose a comprehensive benchmarking framework for earth observation embeddings. The framework specifically aims at evaluating embeddings, foregoing end-to-end finetuning and requiring no access to models used to produce embeddings. The benchmarking procedure also explicitly introduces embedding-dimensionality as a dimension for comparison. The datasets used consist of 5 kinds of tasks with moderate sample sizes (1,000<N<10,000), split between binary classification and regression. The benchmarking framework has been tested in a real-world setting and integrates with at least one established submission platform.
Overall this work presents a solid foundation for EO embedding evaluation but should be expanded to include more evaluation tasks and dataset, and, less urgently, more diverse evaluation settings.

**Strengths:**

- The fundamentals of this submission are sound. The included tasks and scoring procedure are well-motivated, as is the overall work, with current model comparison often not considering embedding dimensionality and how that affects task performance and viability of downstream implementation of the models in any real data pipelines.
- The fact that this framework has already been successfully explored in a practical setting, including integration with an established submission platform, is encouraging and definitely a strength of this work.
- The results on MLP evaluation versus linear evaluation at least somewhat justify the minimal evaluation models.
- The experiments assessing the framework are overall well done and support the framework in its current state.

**Weaknesses:**

- The authors claim in the Introduction the benchmark tasks include “novel EO downstream tasks” but it is unclear what this refers to? Are the datasets novel? Certainly the tasks are based on foundational EO problems.
- The authors repeatedly claim that an issue with embedding models that goes underexplored is the fact that the embeddings they produce exceed the dimensionality of the actual input data, causing issues of data transfer and efficient pipelining. While this is technically true, in practice for ViTs, often only the CLS token is used for downstream tasks while convolutional embeddings are usually pooled into just one spatial dimension. There is still merit to this work and encouraging low embedding dimensionality, but it is unclear how much of a real limitation this is in practice.
- The tasks evaluated seem lacking. It should be straightforward to include at least the subset of the copernicus bench (arXiv:2503.11849) datasets that have the required CC-BY 4.0 license. The authors note this requirement in the Future Work section but without any such effort the benchmark is severely lacking in maturity and ultimately disconnected from central pieces of the EO literature.
- The description of the procedure for equalizing embedding size between the models seems somewhat arbitrary and is hard to understand. Maybe this can be justified further or at least described more clearly.
- Other work shows that evaluating with shallow MLPs can yield different performance estimates from simple linear models. The justification for not including this based on the fact that the MLPs might “compensate for weak embeddings” is inadequate as they have no additional information beyond what is extracted by the embedding model. Especially as more datasets and tasks get included, broadening the evaluation to include at least some non-linear models is critical.

**Questions:**

The scoring is fine but seems sensitive in early stages of the benchmark, when std estimates can fluctuate widely from the impact of a single submission. Have the authors explored any alternative scores to supplement the early phases of releasing this benchmark or simply adding more common embedding methods (even just ImageNet pretrained models or general vision foundation models) to solidify the std estimates. Also, will old scores be updated as new submissions are received and scored?

---

> ### Author Response · Authors · 2025-11-19
>
> We thank the reviewer for the detailed feedback and address each point below. Revised contents we mark green in the updated manuscript. We are going to add results and corresponding discussions by Nov 24, 2025.
>
> 1. __Novelty of Downstream Tasks__
>    All tasks are constructed from curated Sentinel-1/2 registered with corresponding label datasets, making the suite itself novel. Although the tasks reflect common EO problem types, each is composed of new data to avoid any risk of tuning to existing public datasets in a challenge setting. However, we agree that there is ample opportunity to extend the current set by incorporating other existing sources, cf. item 3 below.
>
> 2. __Experiments with CLS Tokens__
>    It is correct that many ViT pipelines rely solely on the CLS token, and that convolutional features are commonly globally pooled. Our baseline experiments already follow these practices: CNN FMs are globally pooled, and ViT FMs are pooled across patches. We also ran CLS-only variants and can include these in the appendix for completeness.
>    Beyond specific FM conventions, our intention is broader: the benchmark provides a standardized environment to systematically study how embedding dimensionality and pooling choices affect performance—an aspect rarely evaluated in a unified manner. While future extensions (as noted in Sec. 6) will include pixel-level tasks, the current framework already enables consistent comparison across FM-based and non-FM embedding methods.
>
> 3. __Inclusion of Copernicus-Bench Downstream Tasks__
>    Copernicus-Bench's peer-reviewed publication dates to Oct 2025---outside the window of required related-work comparisons for the 2026 ICLR paper submission deadline. Nevertheless, the revised version of our manuscript cites Copernicus-Bench. Importantly, our framework is not tied to a particular dataset: any task with embeddings and labels in the expected format (e.g., for classification and regression as simple as a corresponding CSV file) is readily integrated into our framework for evaluation. Incorporating subsets of Copernicus-Bench is therefore a natural avenue for future work by us, the Copernicus-FM/Bench authors, or jointly as a community.
>
> 4. __Embedding Size__
>    We will revise the description of how the input datacube is encoded into compact embeddings for the FM baselines. We apply a fixed temporal aggregation across all models, and fixed spatial aggregation per architecture family (spatial pooling for convolutional encoders, patch pooling for ViT encoders). The updated text will clarify this procedure and make the harmonization of embedding sizes more transparent.
>
> 5. __Non-linear Probing__
>    We agree that non-linear probes are valuable. Section B.3.3 already discusses their role and limitations within the benchmark. Our focus in this release is on linear probes for strict comparability across embedding types. Extending the benchmark with non-linear evaluators is planned future work.
>
> 6. __Additional Baselines__
>    We provide two simple baselines (Teams 10 and 11 in Fig. 7), and additional baselines are now available through open-sourced competition submissions. We will disclose them after de-anonymization in the camera-ready version of our manuscript. Because our approach stores only small embedding files, adding community-contributed baselines to our benchmark framework is straightforward.
>
> 7. __Alternative Scores__
>    Regarding scoring and leaderboard updates: the std-weighted ranking is primarily intended for challenge scenarios, where new submissions shift task difficulty estimates and an overall ranking across multiple tasks is needed. As shown in Fig. 7 (e.g., Team 3’s submission 14), all prior submissions are automatically re-scored upon each new entry. For ongoing non-challenge evaluations, we also report the mean ($Q_t^{(p)}$) and mean per-task metrics (e.g. $R^2$ for regression and $F_1$ for classification), cf. Figs. 4/5/12-14, Tab. 4, which are independent of other methods’ performance and provide a stable alternative. We carved out more clearly these two scenarios in the revised version of our paper.

---

### Official Review · Reviewer_744k · 2025-11-01

**Soundness:** 2
**Presentation:** 1
**Contribution:** 1
**Rating:** 2
**Confidence:** 5

**Summary:**

The paper presents a benchmark for remote sensing embeddings — i.e., the output features of remote sensing foundation models. It appears to be largely a write-up of challenge results from a competition run at the CVPR EarthVision workshop.

The benchmark covers five tasks and six comparison models, which is narrower in scope than recent related efforts such as Pangea or GeoBench. Section 3 provides a clear description of how to set up a dataset challenge and evaluation framework. However, the task selection reveals a strong geographic bias towards the US and Europe (Fig. 2), and the taxonomy of tasks is not clearly motivated in terms of coverage of the remote sensing problem space. For example, there is no marine/water domain task (e.g., marine litter like MADOS), and tasks such as “Landcover Agriculture” and “Crops” seem thematically overlapping. Task difficulty is also not discussed: e.g., crop type classification can be a simple binary setting (“soy vs corn”) or a highly fine-grained setting with hundreds of visually similar classes (e.g. EuroCrops).

The results are also not particularly conclusive. The analysis of embedding size vs. performance is potentially interesting, but the underlying mechanisms are not investigated. For example: how correlated are the embedding dimensions (e.g., how many principal components capture most of the variance)? How does the curse of dimensionality play into these observations? What semantic / spatial / temporal patterns do these embeddings capture?

This paper clearly captures a large amount of work, but, I believe, for ICLR, a stronger contribution would require going beyond ranking models on a challenge, and instead probing and explaining the structure of these embeddings. As written, this reads more like a data challenge report suited to a domain workshop than a venue focused on advancing ML understanding.

**Strengths:**

* The underlying challenge presents extensive work summarizing the results of a benchmarking challenge.
* The description of the benchmarking framework provides a good example how to implement a challenge and outline its main take-aways.

**Weaknesses:**

* Narrow scope compared to existing benchmarks (only 5 tasks / 6 models).
* Strong geographic bias in dataset tasks (mostly US / Europe).
* Task taxonomy and coverage not well justified; missing important domains (e.g. marine) and overlapping task definitions.

**Questions:**

* How relevant is the "compression" aspect to the models and results of this benchmark? How would an embedding from a classic compression algorithm Discrete Cosine Transform (DCT) to create non-deep features for comparison?
* One of the main conclusions was that larger embedding sizes degrade performance, but what are here the underlying factors? The curse of dimensionality, the underlying low dimensional manifold? What are the scaling laws here? It still seems that Terramind is improving with larger embedding sizes?

---

> ### Author Response · Authors · 2025-11-19
>
> We thank the reviewer for their time to evaluate our work. Below we address aspects the review touched upon. Revised contents we mark green in the manuscript.
>
> 1. __Scope of Tasks__
>    While our dataset currently covers five tasks, the central contribution of this work is the benchmarking _framework_ (Fig. 1; App. B.2-B.3) and its metric involved (App. A.1). The provided dataset is intended as a starter resource enabling an end-to-end demonstration of the pipeline rather than an exhaustive coverage of the remote sensing problem space. We agree that additional domains, including marine/water, are valuable and note that a marine use case is already under development for public release.
>
> 2. __Task Difficulty__
>    Our paper does address task difficulty quantitatively via task weights (Fig. 3) and the weighting procedure in Eq. (4). The weighting is one measure to distinguish tasks, e.g., in our setup the landcover tasks have been the most discriminative to rank models. The random (control) task was the most "difficult" as all models failed (by design). From the analysis of results, estimating biomass seems a challenge for real-world downstream tasks. We will add these observations to the revised version of our manuscript.
>
> 3. __Interpretation of Embeddings__
>    Questions regarding correlation structure, principal components, or semantic patterns touch on explainability research, which is beyond the scope of our benchmark framework. Our focus is to establish a reproducible evaluation pipeline and metric, not to probe the internal structure of model embeddings. However, we release the framework to the academic public, and invite extensions that tap into the realm of xAI for enhanced benchmark metrics.
>
> 4. __Relevance of Compression__
>     For now, our benchmark focuses on data "compression" / reduction into fixed-size representations. Correspondingly, the "compression factor" is reflected by the dimension of the embeddings. For example, our $4\times27\times264 \times264\approx7.5M$--dimensional SSL4EO-S12 data cube gets collapsed into a $1024$--dimensional feature vector. As noted in Sect. 6, aspects of entropy-coding we consider future extensions of our work. The current paper does not evaluate (classical) compression algorithms; rather, it provides a framework within which such methods could be incorporated and fairly compared to other methods. We provide some additional thoughts in item 5. below.
>
> 5. __Classical Baselines__
>    We appreciate the suggestion w.r.t. classical compression and baseline methods. Our benchmark already includes a very simple classical baseline---cf. the  channel-wise mean in Figs. 4 and 7---which serves as an example of "compressed, classical non-deep features". Our setup with SSL4EO-S12 data cubes compresses by a factor of more than 7k, i.e., 4 seasons of 27 grayscale images of size 264x264 get compressed into a 1,024-dimensional feature vector. For a DCT it would leave us with about $k_m\propto3$ DCT-components per grayscale image per spatial dimension. This corresponds to an approximate spatial resolution of $10m\cdot264/k_m=880$ meters. Similarely, our mean baseline spatially downsamples to 330 meters in space and averages groups of channels. There are various classical ways of how to compress the SSL4EO-S12 datacube into a 1,024-dimensional feature vector.
>    Please indicate as follow-up to our response that you see value in an additional JPEG-like compression baseline that picks the lowest 9=3x3 components of a DCT per SSL4EO-S12 datacube channel. We are open to conduct additional experiments and add corresponding insights as bonus to the revised version of our manuscript by the end of this year.
>
> 6. __Embedding Size__
>    To avoid a misunderstanding: We do not position degradation at large embedding sizes as a main conclusion, but rather as an empirical observation affecting some--but not all--models and evaluated tasks. For example, TerraMind continues to improve with larger embeddings. A detailed investigation of underlying causes (e.g., linear-probe regime, label volume, manifold assumptions) would be valuable but lies outside the scope of this paper.
>    As we anticipate in Sect. 6, _Future Work_, there exists multiple directions to extend our framework we introduced, and we warmly invite the reviewer and the entire community to join efforts.

---

### Author Response · Authors · 2025-11-25
**summary of updates in manuscript**

We would like to thank all reviewers again for their time and effort put into this evaluation. We updated the paper based on feedback and addressed all comments individually in the reviewer-specific responses last week.

This comment summarizes recent updates to the manuscript that address reviewer feedback which we consider valuable additions. All modifications are highlighted green in the uploaded document, in particular:

1. Clarity on fixed embedding dimension in our general evaluations (cf. reviewer `uuFe`).
2. Additional experiments with GeoFM embeddings (cf. reviewer `vNTR`)
3. Relation of experiments to common embedding practices such as CLS-token usage and spatial averaging (cf. reviewer `uuFe`).

These aspects are important for community adoption of the benchmark framework, and we have strengthened the manuscript accordingly:

- __Methodology update__ (Sect. 5.3, _General Evaluations_): We clarified our embedding aggregation procedures and emphasise that we use simple, consistent choices aligned with common EO practice (spatial averaging, patch/CLS evaluations).
- __Expanded GeoFM baselines__: We now include three additional foundation models (Prithvi, Clay, Satlas) and provide additional CLS-token evaluations. The extended results are shown in Appendix B.3.2 (Table 5) and offer a systematic comparison of baselines.
- __Pre- vs. post-temporal aggregation__: These results are now clearly isolated and presented in Appendix B.3.1, improving the structure of the temporal aggregation analysis and the appendix overall.
- __Additional refinements__:
    * Updates in Sect. 2 _Related Work_ to include Implicit Neural Representations in response to reviewer `vNTR`
    * Clarification on our recommendation of when to use the dynamically updated scoring $s^{(p)}$ and when to use static metrics like the mean quality score $Q_T^{(p)}$ in response to reviewer `uuFe`

---

### Meta-Review · Area_Chair_rVj6 · 2026-01-06

**Summary:**

This submission to the datasets and benchmarks track has presented a 'Earth Observation' (EO) benchmark framework with curated multispectral, multitemporal EO data and an evaluation pipeline with scoring considering both accuracy and stability. The presented work focused on evaluating embedding models in the EO context.

**Reviewer Concerns:**

Based on the available discussions, all reviewers, including reviewer vNTR who gave score 8, expressed the concerns that it may be beneficial to include more embedding baselines. The authors stated that the purpose of the submission is to present this 'reproducible evaluation pipeline' as it is.

Reviewers (especially 744k and uuFe) expressed concerns on tasks with narrow scope, limited coverage, and potential bias. Some also suggested the benchmark framework 'should be expanded to include more evaluation tasks and dataset, and, less urgently, more diverse evaluation settings.' The authors did not seem to provide significant revisions addressing these raised concerns.

**Reviewer Scores:**

Based on the available discussions, it appears that the reviewers may not be able to reach the consensus but all the raised concerns appear to be reasonable. The authors may want to consider addressing these in their future submission.

There were raised ethics concerns during the rebuttal discussion, which may need to be checked as the authors requested.

---

### Decision · Program_Chairs · 2026-01-26

Reject